# PROVABLE CONVERGENCE AND LIMITATIONS OF GEOMETRIC TEMPERING FOR LANGEVIN DYNAMICS

**Omar Chehab, Anna Korba, Austin Stromme & Adrien Vacher**
Department of Statistics
CREST, ENSAE, IP Paris
France
{emir.chehab,anna.korba,austin.stromme,adrien.vacher}@ensae.fr

## ABSTRACT

Geometric tempering is a popular approach to sampling from challenging multi-modal probability distributions by instead sampling from a sequence of distributions which interpolate, using the geometric mean, between an easier proposal distribution and the target distribution. In this paper, we theoretically investigate the soundness of this approach when the sampling algorithm is Langevin dynamics, proving both upper and lower bounds. Our upper bounds are the first analysis in the literature under functional inequalities. They assert the convergence of tempered Langevin in continuous and discrete-time, and their minimization leads to closed-form optimal tempering schedules for some pairs of proposal and target distributions. Our lower bounds demonstrate a simple case where the geometric tempering takes exponential time, and further reveal that the geometric tempering can suffer from poor functional inequalities and slow convergence, even when the target distribution is well-conditioned. Overall, our results indicate that geometric tempering may not help, and can even be harmful for convergence.

## 1 INTRODUCTION

Sampling from a target distribution $\pi$ whose density is known up to a normalizing constant is a challenging problem in statistics and machine learning, and is currently the subject of intense interest due to applications in Bayesian statistics (Dai et al., 2020) and energy-based models in deep learning (Song et al., 2021a), among other areas. In these settings, the normalizing constant of the target distribution $\pi$ is typically intractable, and Markov Chain Monte Carlo (MCMC) algorithms (Roberts and Rosenthal, 2004; Robert and Casella, 2004) are commonly used to generate Markov chains in the ambient space, whose law eventually approximates the target distribution.

Among MCMC algorithms, the Unadjusted Langevin Algorithm (ULA), which corresponds to a time discretization of a Langevin diffusion process, has attracted considerable attention due to its simplicity, theoretical grounding, and utility in practice (Roberts and Tweedie, 1996; Wibisono, 2018; Durmus et al., 2019; Song and Ermon, 2019). For example, ULA can be proven to converge quickly when the target distribution $\pi$ is smooth and strongly log-concave (Durmus and Moulines, 2016; Durmus et al., 2019). However, many cases in practice require to sample from distributions which are not log-concave, and indeed potentially even multi-modal (Parisi, 1981; Zhang et al., 2020). In such settings, the convergence of ULA is governed by functional inequalities which effectively quantify the convexity, or lack thereof, of the target distribution (Vempala and Wibisono, 2019). Nonetheless, truly multi-modal target distributions generally have poor functional inequalities, thus leading to weak convergence guarantees for ULA. This phenomenon is not merely a theoretical artifact, and it is well-known amongst practitioners that when sampling from multi-modal distributions, algorithms based on ULA can get stuck in local modes and suffer from slow convergence (Deng et al., 2020).

*Tempering* or *annealing* is a popular technique (Neal, 1998; Gelman and Meng, 1998; Syed et al., 2022) to overcome the deficiencies of ULA and other MCMC methods in the multi-modal setting. Rather than sample directly from the target distribution $\pi$, tempering samples from a sequence of distributions that interpolate between an easier, unimodal proposal distribution $\nu$ and the more chal-

lenging $\pi$. Intuitively, tempering may help escape local modes and explore the entire target distribution (Syed et al., 2022). Many possible interpolating paths for tempering exist, but to be practically useful the path must be implementable with the chosen MCMC scheme, and should improve convergence when compared the latter run directly against the target $\pi$.

One of the most popular choices for the tempering path is the *geometric tempering* (Gelman and Meng, 1998; Neal, 1998), where the intermediate distributions are defined to be the geometric averages $\mu_\lambda \propto \nu^{1-\lambda}\pi^\lambda$ with $\lambda \in [0, 1]$ the (inverse) temperature parameter. At a "hot" temperature, $\lambda$ is close to 0 and the distribution $\mu_\lambda$ is close to the proposal which can be chosen with large variance for better exploration, and when the temperature is gradually "cooled" to $\lambda = 1$, $\mu_\lambda$ recovers the original target $\pi$. In practice, the geometric tempering does not require access to the normalizing constants, and has accordingly been widely applied to various MCMC methods (Dai et al., 2020; Chopin and Papaspiliopoulos, 2020). However, theoretical understanding of the geometric tempering is limited. This includes identifying situations where it improves upon standard sampling procedures as well as offering guidance for the selection of the temperature schedule, a crucial question in practice where a number of heuristics have been proposed (Chopin and Papaspiliopoulos, 2020; Jasra et al., 2011; Chopin et al., 2024; Kiwaki, 2015).

**Contributions.** In this paper, we develop theoretical understanding of geometric tempering combined with Langevin dynamics: we refer to this as tempered Langevin dynamics. We make three main contributions:

1. We provide in Theorems 1 and 3 the first convergence result for tempered Langevin dynamics in Kullback-Leibler divergence (KL). Our results here are for general tempering schedules and depend on the functional inequalities (log-Sobolev) of the intermediate distributions along the tempering sequence. In Proposition 6 we derive the optimal tempering schedule for our continuous time upper bound in the strongly log-concave setting.

2. To go beyond the strongly-log concave setting, we must understand the behavior of functional inequalities along the geometric tempering. Our result here, Theorem 4, is negative, and shows that, surprisingly, even when the proposal and target have favorable functional inequalities, the geometric tempering can exponentially worsen these inequalities.

3. Although the poor functional inequalities in Theorem 4 are a worrying sign for the geometric tempering, they do not yet rule out fast convergence since functional inequalities only govern worst-case convergence. We thus analyze a simple bi-modal example where we show in Theorem 8 that the geometric path takes exponential time to converge in total variation (TV), and then show in Theorem 9 that, surprisingly, similar results even hold for a uni-modal target with favorable functional inequalities.

In sum, our results establish sufficient conditions for the tempered Langevin dynamics to converge in KL divergence. We find some limited situations where these improve upon the rates of standard Langevin dynamics, but we also find that the geometric tempering can worsen functional inequalities and suffer from slow convergence, both in the setting of multi-modal target and even for uni-modal targets with reasonable functional inequalities.

**Organization.** This paper is organized as follows. In the remainder of this section we discuss related work and describe our notation. In section 2, we discuss background about ULA and geometric tempering. In section 3, we state our upper bounds in KL on the convergence of the geometric tempering with Langevin dynamics under functional inequalities, both in continuous and discrete time. In section 4, we show that the geometric path can have poor functional inequalities even when the proposal and target do not; yet, in the strongly-concave case, we derive explicit results from our upper bounds and highlight situations where tempering may be beneficial. In section 5, we describe two examples where geometric tempering with ULA provably has slow convergence in TV. Proofs and additional numerical validations are collected in the Appendix.

**Related work.** Given the empirical success of ULA with a sequence of tempered target distributions it is desirable to obtain theoretical guarantees on the convergence of the scheme, and especially to understand when and why certain sequences of moving targets can guide or misguide the sampling process towards the final target. Closely related to our setting, Tang et al. (2024) also study the convergence of tempered Langevin dynamics for the geometric sequence. However, they focus on the

simulated annealing setting. In this case the geometric sequence is taken as $\pi^\lambda$, that degenerates to a target distribution which is a Dirac located at the global maximum of the log density of $\pi$ as $\lambda$ goes to infinity, so that sampling actually becomes an optimization task (Kirkpatrick et al., 1983; Cerný, 1985). Using an explicit temperature schedule, they measure convergence in probability. For another sequence of intermediate targets obtained by convolving the target and proposal distributions, and using an explicit schedule, Lee et al. (2022) prove fast convergence in TV. Finally, for a general sequence of intermediate targets, Guo et al. (2024) very recently obtained a rate of convergence that depends on Wasserstein metric derivative along the path of distributions: their result suggests that an optimal path can be obtained as a Wasserstein geodesic between the initial and target distributions. A modification of tempered Langevin dynamics, called Sequential Monte Carlo, that importantly includes resampling of particles, has been shown to achieve fast convergence in TV (Schweizer, 2012; Paulin et al., 2018; Mathews and Schmidler, 2024; Lee and Santana-Gijzen, 2024). These results apply to different sampling processes and rely on strong assumptions, effectively modeling far-away modes by disjoint sets (Schweizer, 2012; Mathews and Schmidler, 2024), using a specific path that interpolates between a uniform and target distribution by increasing the number of components that follow the target's law (Paulin et al., 2018), or assuming uniformly bounded consecutive distributions along the path (Lee and Santana-Gijzen, 2024). In contrast, our results are specific to the geometric sequence, but we obtain upper and lower bounds on convergence that explicitly depend on the time.

The tempered iterates defined by the geometric path are also at the basis of simulated or parallel tempering schemes (Geyer, 1991; Marinari and Parisi, 1992; Hukushima and Nemoto, 1996; Syed et al., 2021), which are MCMC algorithms where the temperature is a random variable instead of a monotonic function of time. Both schemes produce samples at all temperatures and involve swapping particles between hotter and colder temperatures. In that setting, some works have investigated the spectral gap of these methods, which is related to the rate of convergence in TV distance (Madras and Zheng, 2003; Woodard et al., 2009a;b). When that rate is polynomially (resp. exponentially) decreasing in the problem size, convergence is said to be fast (resp. 'torpid'). These rates are studied for arbitrary target distributions, using lower and upper bounds. Namely, Woodard et al. (2009a;b) show that for target distributions which have modes with different weights or shapes, convergence can be slow, and that symmetric modes are required for fast convergence. This generalizes previous findings that for specific targets with two symmetric and equally weighted modes, convergence is fast (Madras and Zheng, 2003), whereas for another specific target with three asymmetric modes, convergence is slow for any schedule (Bhatnagar and Randall, 2015). Ge et al. (2018) also prove fast convergence when the target is a Gaussian mixture. More recently, Chen et al. (2019) study the Poincaré constant in parallel tempering, which governs the rate of convergence in $\chi^2$ divergence: they show it improves upon standard Langevin, and relate this improvement to the exchange rate of particles between different temperatures. Our work differs in several important ways: we study a different sampling algorithm (Unadjusted Langevin algorithm), obtain upper and lower bounds directly in time rather than on the spectral gap, and prove explicit rates of convergence, as well as lower bounds, for simple choices of proposal and target distributions.

**Notation.** $\mathcal{C}_c^\infty(\mathbb{R}^d)$ denotes the set of infinitely differentiable functions with compact support. $\mathcal{P}(\mathbb{R}^d)$ denotes the set of probability measures $p$ on $\mathbb{R}^d$. For $p \in \mathcal{P}(\mathbb{R}^d)$, we denote that $p$ is absolutely continuous w.r.t. $q$ using $p \ll q$ and we use $dp/dq$ to denote the Radon-Nikodym derivative. The set of probability measures which are absolutely continuous with respect to the Lebesgue measure is written $\mathcal{P}_{\mathrm{ac}}(\mathbb{R}^d)$. The Total Variation (TV) distance is defined as $\mathrm{TV}(p,q) := \sup_{A \subset \mathbb{R}^d} |p(A) - q(A)|$, where the supremum runs over all Borel sets. For any $p \in \mathcal{P}(\mathbb{R}^d)$, $L^2(p)$ is the space of functions $f : \mathbb{R}^d \to \mathbb{R}$ such that $\int \|f\|^2 dp < \infty$. We denote by $\|\cdot\|_{L^2(p)}$ and $\langle \cdot, \cdot \rangle_{L^2(p)}$ respectively the norm and the inner product of the Hilbert space $L^2(p)$. For $p \ll q$, the Kullback-Leibler (KL) divergence is defined as $\mathrm{KL}(p,q) = \int \log\left(\frac{dp}{dq}\right) dp$, the $\chi^2$-divergence as $\chi^2(p,q) := \int \left(\frac{dp}{dq} - 1\right)^2 dq$ and the Fisher-divergence as $\mathrm{FD}(p,q) = \left\|\nabla \log\left(\frac{dp}{dq}\right)\right\|_{L^2(p)}^2$, and $+\infty$ otherwise. For a measurable function $f : \mathbb{R}^d \to \mathbb{R}$ we define the variance $\mathrm{Var}_p(f) := \mathbb{E}_p[(f - \mathbb{E}_p[f])^2]$ for $p \in \mathcal{P}(\mathbb{R}^d)$. We write the standard Gaussian on $\mathbb{R}$ with mean $a$ and variance $\sigma^2$ as $\mathcal{N}(a, \sigma^2)$, and the uniform measure on a Borel set $A \subset \mathbb{R}^d$ with finite and positive Lebesgue measure as $\mathrm{unif}_A$. Given a closed set $A \subset \mathbb{R}^d$ we define $d(x, A) := \inf_{y \in A} \|x - y\|$ for all $x \in \mathbb{R}^d$. Finally, we define the error function by $\mathrm{erf}(z) := \frac{2}{\sqrt{\pi}} \int_0^z e^{-x^2} \mathrm{d}x$.

## 2 BACKGROUND

In this section, we provide some background on functional inequalities, Langevin dynamics, and geometric tempering.

**Functional inequalities.** Let $q \in \mathcal{P}_{\mathrm{ac}}(\mathbb{R}^d)$. We say that $q$ satisfies the *Poincaré inequality* with constant $C_P \geq 0$ if for all $f \in \mathcal{C}_c^\infty(\mathbb{R}^d)$,

$$\mathrm{Var}_q(f) \leq C_P \|\nabla f\|_{L^2(q)}^2, \tag{1}$$

and let $C_P(q)$ be the best constant in Eq. 1, or $+\infty$ if it does not exist. We say that $q$ satisfies the *log-Sobolev inequality* with constant $C_{LS}$ if for all $f \in \mathcal{C}_c^\infty(\mathbb{R}^d)$,

$$\mathrm{ent}_q(f^2) := \mathbb{E}_q\Big[f^2 \ln\Big(\frac{f^2}{\mathbb{E}_q[f^2]}\Big)\Big] \leq 2C_{LS}\|\nabla f\|_{L^2(q)}^2, \tag{2}$$

and let $C_{LS}(q)$ be the best constant in Eq. 2, or $+\infty$ if it does not exist. Note that log-Sobolev implies Poincaré with the same constant (Bakry et al., 2014), so that $C_{LS}(q) \geq C_P(q)$. If $q \propto e^{-V}$ and the potential $V$ is $\alpha_q$-strongly convex, then $q$ satisfies Eq. 2 with constant $\frac{1}{\alpha_q}$. However, the latter is more general, including for instance distributions $q$ whose potentials are bounded perturbations of a strongly convex potential (Bakry et al., 2014; Cattiaux and Guillin, 2022).

**Langevin dynamics.** Let $\pi \in \mathcal{P}_{\mathrm{ac}}(\mathbb{R}^d)$. The Unadjusted Langevin Algorithm (ULA) (Parisi, 1981; Besag, 1994) consists in sampling a target distribution $\pi$ using noisy gradient ascent

$$X_{k+1} = X_k + h\nabla \log \pi(X_k) + \sqrt{2h}\epsilon_k, \quad \epsilon_k \sim \mathcal{N}(0, \mathrm{I}_d) \tag{3}$$

with step size $h > 0$ at iteration $k \in \mathbb{N}$. Setting time as $t = hk$ and taking the continuous limit obtained as $h \to 0$, Eq. 3 defines a continuous process known as the Langevin diffusion. The convergence of the law of the Langevin diffusion to the equilibrium measure $\pi$ is then governed by the Poincaré and log-Sobolev inequalities. In particular, if we denote the law of the Langevin diffusion at time $t$ by $p_t$, then $p_t$ converges to $\pi$ in KL with exponential rate determined by the log-Sobolev constant of $\pi$ (Vempala and Wibisono, 2019, Theorem 2), namely

$$\mathrm{KL}(p_t, \pi) \leq e^{-2C_{LS}(\pi)^{-1}t} \mathrm{KL}(p_0, \pi). \tag{4}$$

However, for multi-modal distributions such as Gaussian mixtures, the log-Sobolev constant can grow exponentially with the distance between modes (Chen et al., 2021). Nevertheless, ULA remains a popular choice, due to its computational simplicity: simulating Eq. 3 only requires access to the score $\nabla \log \pi$ which does not depend on the target's normalizing constant.

**Langevin with moving targets.** Many heuristics broadly known as annealing or tempering consist in using ULA to sample a path, or sequence of distributions $(\mu_t)_{t \in \mathbb{R}_+}$ instead of the single target $\pi$. The hope is that this sequence of intermediate distributions will improve the convergence of the ULA sampler. Different tempering algorithms sample the path sequentially in time (Dai et al., 2020; Neal, 1998; Rubin, 1987), back-and-forth in time (Lee et al., 2021; Neal, 1996; Zhang et al., 2020), or at all times jointly (Marinari and Parisi, 1992; Geyer, 1991). This paper deals with the first case, i.e.,

$$X_{k+1} = X_k + h_k\nabla \log \mu_k(X_k) + \sqrt{2h_k}\epsilon_k, \quad \epsilon_k \sim \mathcal{N}(0, 1) \tag{5}$$

where the target now is updated ("moved") at each iteration. This generic sampling method has been used in high-dimensional spaces (Wu et al., 2020; Thin et al., 2021; Geffner and Domke, 2023; Song and Ermon, 2019) and has achieved state-of-the-art results for sampling images, where it is sometimes known by the names Annealed Langevin Dynamics (Song and Ermon, 2019) or the "corrector" sampler (Song et al., 2021b). It is therefore of interest to find moving targets whose geometry is well-suited to ULA's convergence properties. A number of paths $(\mu_t)_{t \in \mathbb{R}_+}$ can be used to guide the process toward the final target $\pi$. Many of them interpolate between a proposal distribution $\nu$ that is easy to sample and the target distribution $\pi$, for example by taking their convolution (Song and Ermon, 2019; Song et al., 2021b; Albergo et al., 2023), their geometric mean (Neal, 1998), or following the gradient flow of a loss from proposal to target (Tieleman, 2008; Carbone et al., 2024;

Marion et al., 2024). The path obtained by convolving the two distributions is the default choice for sampling from so-called "diffusion models", yet the scores $\nabla \log \mu_t$ along that path are not analytically tractable in our setting when the density of $\pi$ is known up to a normalization constant, and estimating them is the subject of current research (Huang et al., 2024; He et al., 2024; Grenioux et al., 2024; Saremi et al., 2024).

**Geometric tempering.** The path obtained by taking the geometric mean of the proposal and target distributions has distinguished itself in the sampling literature (Neal, 1998; Gelman and Meng, 1998). It is written as

$$\mu_t(x) = c_{\lambda_t} \nu(x)^{1-\lambda_t} \pi(x)^{\lambda_t}, \quad t \in \mathbb{R}_+, \tag{6}$$

where $c_{\lambda_t}$ is a normalizing factor, $\nu \in \mathcal{P}_{ac}(\mathbb{R}^d)$ is a proposal distribution and $\lambda(\cdot): \mathbb{R}_+ \to [0,1]$ is an increasing function called the tempering schedule. It has recently been shown that the geometric path can be identified to a time-discretized gradient flow of the Kullback-Leibler divergence to the target, with respect to the Fisher-Rao distance (Chopin et al., 2024; Domingo-Enrich and Pooladian, 2023). Its main advantage is its computational tractability, since the score of $\nabla \log \mu_t = (1 - \lambda_t)\nabla \log \nu + \lambda_t \nabla \log \pi$ is known in closed-form. This path is a default in some sampling libraries (Cabezas et al., 2023) and remains a popular choice in recent sampling literature (Thin et al., 2021; Geffner and Domke, 2023; Dai et al., 2020) and applications (Bradley and Nakkiran, 2024; Ramesh et al., 2022; Saharia et al., 2022; Dieleman, 2022). A special case of the geometric path is especially popular, choosing $\nu$ to be "uniform", or more formally, equal to the Lebesgue measure: $\mu_t(x) = c_t \pi(x)^{\lambda_t}$; this choice is commonly used in practice to sample from un-normalized distributions parameterized by a deep neural network (Wu et al., 2020; Grathwohl et al., 2020; Nijkamp et al., 2019; Ye et al., 2017) or for global optimization (Marinari and Parisi, 1992). Note that when $\lambda_0 > 0$, we can define the probability measure $\nu \propto \pi^{\lambda_0}$ and a tempering schedule $\gamma_t := \frac{\lambda_t - \lambda_0}{1 - \lambda_0}$, and rewrite $\mu_t(x) = c_t \pi(x)^{\lambda_t}$ as a special case of Eq. 6 with the tempering schedule $\gamma_t$. When $\nu$ is not chosen to be "uniform", it is often chosen as a simple distribution, such as a Gaussian (Cabezas et al., 2023; Zhang et al., 2021; Thin et al., 2021).

## 3 CONVERGENCE RATE FOR TEMPERED LANGEVIN DYNAMICS

Throughout, we take as given proposal and target distributions $\nu$ and $\pi$, as well as a temperature schedule $\lambda: \mathbb{R}_+ \to [0,1]$, which we assume satisfy the following conditions.

**Assumption 1** (Regularity of proposal, target, and tempering) *The proposal $\nu$ and the target $\pi$ have densities with respect to the Lebesgue measure, which we write $\nu \propto e^{-V_\nu}$ and $\pi \propto e^{-V_\pi}$. The tempering schedule $(\lambda_t)_{t \geq 0}$ is such that $\lambda: \mathbb{R}_+ \to [0,1]$ and $\lambda_t$ is non-decreasing in $t$ and weakly differentiable.*

In addition to this basic regularity, we also make the following quantitative assumptions on the negative log densities.

**Assumption 2** (Lipschitz gradients and dissipativity) *The negative log densities $V_\nu$, $V_\pi$ have Lipschitz continuous gradients on all of $\mathbb{R}^d$, with Lipschitz constants $L_\nu, L_\pi$, respectively. In addition, $V_\nu$ and $V_\pi$ satisfy the dissipativity inequalities*

$$\langle \nabla V_\nu(x), x \rangle \geq a_\nu \|x\|^2 - b_\nu, \qquad \langle \nabla V_\pi(x), x \rangle \geq a_\pi \|x\|^2 - b_\pi, \tag{7}$$

*with constants $a_\nu, a_\pi, b_\nu, b_\pi > 0$.*

The Lipschitz assumption is used both for our discrete time results as well as in guaranteeing existence and uniqueness of the continuous time dynamics; see Appendix A.1 for more discussion on this latter point. The dissipativity condition is common in the sampling literature (Conforti, 2024; Erdogdu et al., 2022) as it implies a finite log-Sobolev constant under Lipschitz gradients (Cattiaux et al., 2010). Our only quantitative use of the dissipativity condition will be to get control on the second moments of the tempering dynamics, but it is also convenient to ensure the log-Sobolev constants remain finite along the tempering path. Indeed, dissipativity of both $V_\nu$ and $V_\pi$ implies dissipativity of $\mu_t$ from Eq. 6, so $C_{LS}(\mu_t) < \infty$. Our results will crucially rely on the size of the inverse of these log-Sobolev constants, so we define the notation

$$\alpha_t := C_{LS}(\mu_t)^{-1} > 0, \qquad \forall t \geq 0. \tag{8}$$

As a simple example, notice that if $V_\nu, V_\pi$ are $\alpha_\nu, \alpha_\pi$ strongly convex, then $\alpha_t \geq (1-t)\alpha_\nu + t\alpha_\pi$. In section 4.1, we investigate how $\alpha_t$ depends on the log-Sobolev constants of $\nu$ and $\pi$, and here proceed with the statement of our convergence results.

Given the proposal $\nu$ and target $\pi$, as well as a tempering schedule $(\lambda_t)_{t\geq 0}$, we are interested in the tempered Langevin dynamics, where the moving target is the geometric path $\mu_t$, as defined in Eq. 6. In continuous time, this is the stochastic differential equation

$$dX_t = -\big((1-\lambda_t)\nabla V_\nu + \lambda_t \nabla V_\pi\big)dt + \sqrt{2}dW_t, \tag{9}$$

with initialization $X_0 \sim p_0$ for some $p_0 \in \mathcal{P}(\mathbb{R}^d)$, and where $W_t$ is a standard $d$-dimensional Brownian motion. We denote by $p_t$ the density of $X_t$ given by the continuous time dynamics Eq. 9. Our main upper bound in continuous time follows.

**Theorem 1** (Continuous time) *Suppose Assumption 1 and 2 hold, and let $(\alpha_t)_{t\geq 0}$ be the inverse log-Sobolev constants as in Eq. 8, assumed to be integrable. Let $p_t$ be the law of Eq. 9 with initialization $p_0$ and denote by $\dot\lambda_t$ the weak time derivative of the tempering schedule. Then, for all $t \geq 0$:*

$$\mathrm{KL}(p_t, \pi) \leq \exp\left(-2\int_0^t \alpha_s ds\right)\mathrm{KL}(p_0, \mu_0) + A(1-\lambda_t) + A\int_0^t \dot\lambda_s \exp\left(-2\int_s^t \alpha_v dv\right)ds \tag{10}$$

*where $A = 2(L_\pi + L_\nu)(\frac{2(d+b_\nu+b_\pi)}{a_\nu \wedge a_\pi} + \mathbb{E}_{p_0}[\|x\|^2])$ depends on the constants from Assumption 2, the second moment of the initial distribution $p_0$, and linearly on the dimension $d$.*

The proof of Theorem 1 can be found in Appendix A.2. To the best of our knowledge, Theorem 1 is the first convergence analysis of Tempered Langevin dynamics, namely Eq. 9, in the literature. The first two terms of the upper bound deal with the start and end of the tempering: the first one measures the convergence rate to the *first* tempered distribution $\mu_0$ and the second one measures how far the *current* tempered distribution is from the target. The first term is set to zero when the schedule starts with $\lambda_t = 0$ and the sampling process Eq. 9 is initialized with the proposal $p_0 = \nu$. The second term is null when $\lambda_t = 1$ at the time when convergence is evaluated. The last term in the upper bound involves the tempering speed $\dot\lambda$, as well as the geometry of the moving targets $\mu_t$ via their inverse log-Sobolev constants $\alpha_t$ between inverse temperatures $\lambda_0$ and $\lambda_t$, and, in particular, is null when the annealing schedule is constant. Note that, in Appendix A.3, we translate Theorem 1 to give sufficient conditions for convergence to the target at precision $\mathrm{KL}(p_t, \pi) < \epsilon$.

**Remark 2** (Recovering standard upper bound for Langevin dynamics without tempering) *Notice that Eq. 9 recovers standard Langevin dynamics when we set $\lambda_t \equiv 1$ for all $t \in \mathbb{R}_+$. In this case, only the first term in the upper bound is non-zero, and the bound recovers the standard continuous-time upper bound for Langevin dynamics, as recalled in Eq. 4.*

Next, we analyze the Euler-Maruyama discretization of the continuous time Tempered Langevin Dynamics Eq. 9. Namely, we take $X_0 \sim p_0$ and then, for a tempering sequence $(\lambda_k)_{k\geq 0}$ and a sequence of step-sizes $(h_k)_{k\geq 0}$, we follow the iteration

$$X_{k+1} = X_k - h_{k+1}\big((1-\lambda_{k+1})\nabla V_\nu + \lambda_{k+1}\nabla V_\pi\big) + \sqrt{2h_{k+1}}\epsilon_{k+1}, \tag{11}$$

where $\epsilon_{k+1}$ is independent standard Gaussian noise. In other words, at each iteration $k$ we take a Langevin step of size $h_{k+1}$ towards $\mu_{k+1} := c_{k+1}\nu^{1-\lambda_{k+1}}\pi^{\lambda_{k+1}}$.

**Theorem 3** (Discrete time) *Suppose Assumptions 1 and 2 hold, and let $(\alpha_k)_{k\geq 1}$ be the inverse log-Sobolev constants as in Eq. 8. Define the smoothness constant of the tempered path to be $L_k := (1-\lambda_k)L_\nu + \lambda_k L_\pi$, and let $p_k$ be the law of Eq. 5 with initialization $p_0$. Then, so long as*

$$h_k \leq \min\left(\frac{\alpha_k}{4L_k^2}, \frac{a_\pi \wedge a_\nu}{2(L_\pi + L_\nu)^2}, 1\right)$$

*for all $k$, we have*

$$\mathrm{KL}(p_k, \pi) \leq \exp\left(-\sum_{j=1}^k \alpha_j h_j\right)\mathrm{KL}(p_0, \mu_0) + A'(1-\lambda_k)$$

$$+ A'\sum_{i=1}^k (\lambda_i - \lambda_{i-1})\exp\left(-\sum_{j=i}^k \alpha_j h_j\right) + 6\sum_{i=1}^k h_i^2 dL_i^2 \exp\left(-\sum_{j=i+1}^k \alpha_j h_j\right) \tag{12}$$

*where $A' = 2(L_\pi + L_\nu)\big(\max\big(\mathbb{E}_{p_0}[\|x\|^2], \frac{2(3(b_\pi+b_\nu)/2+d)}{a_\pi \wedge a_\nu \wedge 1}\big) + \frac{3(d+b_\nu+b_\pi)}{a_\pi \wedge a_\nu}\big)$ depends on the constants from Assumption 2, the second moment of the initial distribution $p_0$, and linearly on the dimensional d.*

The proof of Theorem 3 can be found in Appendix B. The first three terms are the discrete-time equivalent of those obtained in the continuous-time setup of Theorem 1. The novelty is the fourth term which is the bias from the discretization: it becomes null as the step sizes $h_k$ tend to zero. Otherwise, it involves the geometry of the intermediate target distributions via their smoothness constants $L_k$, additionally to their inverse log-Sobolev constants $\alpha_k$. Again, when there is no tempering, i.e. setting $\lambda_k \equiv 1$, since the second and third term cancel, we recover the known upper bound on the convergence of ULA (Vempala and Wibisono, 2019, Th. 2) up to a multiplicative constant. Similarly to the continuous case, we derive in Appendix B.2 sufficient conditions for converging to the target with a given precision $\mathrm{KL}(p_k, \pi) < \epsilon$.

Understanding the upper bounds in this section mainly involves two key points: the geometry of the moving targets via their inverse log-Sobolev constants $\alpha_t$, and the tempering schedule $\lambda(\cdot)$. These will be the focus of the next sections.

## 4 ANALYSIS AND OPTIMIZATION OF THE UPPER-BOUNDS

In this section we explore the continuous-time upper bound from Theorem 1. We first present in section 4.1 a simple example where the log-Sobolev constants of the intermediate distributions along the geometric mean path can be exponentially worse than those of the target and proposal. Motivated by this result, we then conduct a detailed study of the optimal tempering schedule in section 4.2 in the setting where both $\nu$ and $\pi$ are strongly log-concave.

### 4.1 GEOMETRIC TEMPERING CAN EXPONENTIALLY WORSEN FUNCTIONAL INEQUALITIES

Because of the fundamental role that log-Sobolev constants play in governing the convergence of Langevin dynamics without tempering, it is natural that our upper bounds for tempered Langevin dynamics depend on the log-Sobolev constants of the intermediate distributions. Nonetheless, the size of these log-Sobolev constants is crucial for ensuring rapid convergence of tempered Langevin dynamics. We are thus led to ask a fascinating yet, to the best of our knowledge, new question: how do the log-Sobolev constants along the geometric tempering path depend on the proposal and target distributions?

Developing a complete answer to this question is an interesting direction for future work, but here we provide a simple example which shows that, in general, the geometric path can actually make functional inequalities *exponentially worse*. As is commonly done in practice, we take the proposal distribution $\nu$ to be Gaussian (Cabezas et al., 2023; Zhang et al., 2021; Thin et al., 2021). For the target, we take a parameter $m > 0$ and put

$$\pi := (1 - e^{-m^2/4})\mathcal{N}(m, 1) + e^{-m^2/4}u_m, \quad (13)$$

where $u_m$ is the smoothed uniform distribution on $I_m := [-m, 2m]$ with density proportional to $e^{-\frac{1}{2}d(x, I_m)^2}$. Note that without the small mixing with the smoothed uniform distribution in Eq. 13, the target $\pi$ would be strongly log-concave, and the

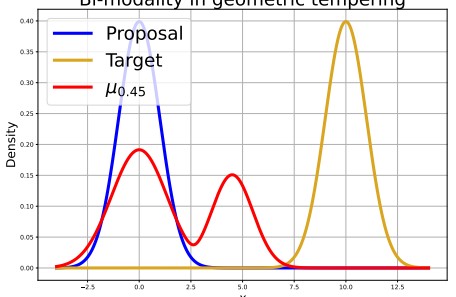

Figure 1: Bi-modal intermediate distribution at $\lambda = 0.45$ with $\nu, \pi$ as in Eq. 13 for $m = 10$.

geometric path would remain well-conditioned. The next result shows that while this small mixing does not greatly worsen the log-Sobolev constant of $\nu$, it does cause the geometric tempering to have exponentially worse log-Sobolev constant, in fact even Poincaré constant.

**Theorem 4** *Let $\nu := \mathcal{N}(0, 1)$ and $\pi$ be as defined in Eq. 13 for $m \geq 10$. Then $C_{LS}(\nu) = 1$ and $C_{LS}(\pi) \leq 81m^5$, yet for all $\lambda \in [\frac{1}{2}, 1]$,*

$$C_P(\mu_\lambda) \geq \frac{1}{4 \cdot 10^4 m} e^{m^2(1-\lambda)/100} - m^2.$$

*Since $C_{LS}(\cdot) \geq C_P(\cdot)$, the above holds, in particular, with $C_{LS}(\mu_\lambda)$ on the left-hand side.*

The proof of Theorem 4 can be found in Appendix D.3. This example is plotted in Figure 1; the intuition is that while both distributions are unimodal and thus well-conditioned, the small mixing with the uniform measure creates multi-modality in the geometric tempering. The proof relies on careful analysis to characterize this multi-modality, and then uses a test function which simply linearly interpolates between the modes to witness the exponentially large Poincaré constant.

Theorem 4 rules out the possibility that the geometric tempering generically improves, or even preserves, functional inequalities. Since our upper bounds depend on the log-Sobolev constants of the geometric tempering, they also can suffer from such exponentially poor conditioning. But since log-Sobolev inequalities only govern worst-case convergence, it is still at least *a priori* possible that geometric tempering nonetheless achieves fast convergence. We investigate this question further in section 5, and here instead continue exploring our upper bounds.

## 4.2 OPTIMAL TEMPERING IN THE STRONGLY LOG-CONCAVE CASE

Other than the log-Sobolev constants $\alpha_t$, the main input to our upper bounds is the tempering schedule $\lambda$, and in this section we use our upper bounds to investigate the optimal schedule, an important question in practice (Grosse et al., 2013; Song and Ermon, 2020). Both to make our analysis tractable and because of the degeneracy in the non log-concave case identified in the previous section, we here restrict to the strongly log-concave setting. Specifically, we assume that both the proposal $\nu$ and the target $\pi$ are log-concave with strong-concavity parameters $\alpha_\pi, \alpha_\nu > 0$. In this case, $\alpha_t \geq (1 - \lambda_t)\alpha_\nu + \lambda_t\alpha_\pi$.

We start by reformulating the continuous time result that we obtained in Theorem 1, in the case where both the proposal $\nu$ and target $\pi$ are log-concave.

**Corollary 5** *Suppose $\nu$ and $\pi$ satisfy Assumptions 1 and 2, are $\alpha_\nu$ and $\alpha_\pi$-strongly log-concave respectively so that $\alpha_t \geq \lambda_t\alpha_\pi + (1 - \lambda_t)\alpha_\nu$, and that the process is initialized at the proposal distribution $p_0 = \nu$. Then for all $t \geq 0$, we have*

$$\mathrm{KL}(p_t, \pi) \leq AG_t(\lambda), \qquad G_t(\lambda) := 1 - 2\int_0^t \lambda_s\alpha_s \exp\big(-2\int_s^t \alpha_v \mathrm{d}v\big)\mathrm{d}s, \qquad (14)$$

*where $A$ is as in Theorem 1. Suppose additionally that $\alpha_\pi \geq \alpha_\nu$. Then $G_t(\lambda)$ is minimized by the vanilla Langevin scheme $\lambda(t) \equiv 1$.*

The proof of Corollary 5 is in Appendix C.1 and relies on integration by parts. This new expression allows us to optimize on the schedule $\lambda$ independently of the *unknown* quantity $A$. As one could expect, the corollary above states that when the target $\pi$ is already better conditioned than the proposal $\nu$, so $\alpha_\pi \geq \alpha_\nu$, there is no need to temper and the optimal tempering scheme is given by vanilla Langevin $\lambda \equiv 1$. In the next proposition, we show on the contrary that if $\pi$ is too poorly conditioned with respect to $\nu$, then one should indeed use a custom tempering scheme other than Langevin.

**Proposition 6** *Suppose $\alpha_\pi < \alpha_\nu$. Then the functional $G_t(\lambda)$ is minimized for the scheme*

$$\lambda(s) = \min\left(\frac{\alpha_\nu}{\alpha_\nu - \alpha_\pi}\frac{1 + \alpha_\nu s}{2 + \alpha_\nu s}, 1\right). \qquad (15)$$

*In particular, the optimal schedule does not depend on the horizon $t$, and when $\alpha_\pi \geq \alpha_\nu/2$, vanilla Langevin $\lambda(t) \equiv 1$ is optimal.*

Proposition 6 is proven in Appendix C.2. Hence when the target is sufficiently peakier than the proposal $\alpha_\pi \geq \alpha_\nu/2$, tempering does *not* improve convergence beyond that of vanilla Langevin. Conversely, when the target is flat enough $\alpha_\pi < \alpha_\nu/2$, then tempered Langevin with the schedule in Eq. 15 *does* improve convergence over vanilla Langevin. While the optimal schedule is provided analytically in Eq. 15, it cannot be computed exactly in practice given that the constant $\alpha_\pi$ that describes the geometry of the target distribution is unknown. On the other hand, in the limit of a flat and log-concave target $\alpha_\pi \to 0$, the optimal schedule tends to $\lambda(s) = 1 - \frac{1}{2 + \alpha_\nu s}$, which can be implemented in practice.

A natural question follows: can other schedules, which may not be optimal but are feasible to implement in general, actually improve convergence beyond that of vanilla Langevin? A simple

example is the linear schedule. Here, we obtain an explicit convergence rate that does not improve on standard Langevin at large times, but that is faster than standard Langevin at small times for small $\alpha_\pi$.

**Proposition 7** *Assume $\alpha_\nu > \alpha_\pi$. Let $t > 0$ until which the continuous process 9 is run. Then, with the linear tempering scheme $\lambda(s) \equiv \frac{s}{t}$ defined on [0,t], $G_t$ in the upper-bound of Corollary 5 writes*

$$G_t(\lambda) = \sqrt{\frac{\pi}{4t(\alpha_\nu - \alpha_\pi)}} \left\{ \mathrm{erfcx}\left(\alpha_\pi \sqrt{\frac{t}{\alpha_\nu - \alpha_\pi}}\right) - e^{-(\alpha_\nu + \alpha_\pi)t} \, \mathrm{erfcx}\left(\alpha_\nu \sqrt{\frac{t}{\alpha_\nu - \alpha_\pi}}\right) \right\},$$

*where* $\mathrm{erfcx}$ *is the complementary scaled error function* $\mathrm{erfcx}(x) = e^{x^2}(1 - \mathrm{erf}(x))$. *In particular, as $t \to \infty$, we have $G_t(\lambda) \sim \frac{1}{2\alpha_\pi t}$.*

The proof of Proposition 7 appears in Appendix C.3.

Finally, we compare in Figure 2 the value of the upper-bound $G$ at different time horizons $t$ using the optimal tempering scheme given in Eq. 15, the linear tempering scheme $\lambda(s) \equiv \frac{s}{t}$ and vanilla Langevin $\lambda(s) \equiv 1$, for $\alpha_\pi = 0.01$ and $\alpha_\nu = 1.0$. We observe that the optimal schedule always yields a lower value of $G$ (as it should) and that it provides a clear advantage over vanilla Langevin at short horizons $t$. However, as the horizon grows, this edge is eventually lost. Similarly, we observe that the linear tempering schedule improves over vanilla Langevin at short time horizons yet is eventually beaten as the horizon grows.

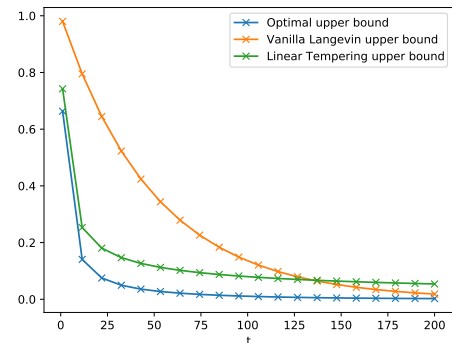

Figure 2: Value of the upper-bound $G$ uling the optimal tempering, linear tempering, and standard Langevin.

## 5 EXPONENTIALLY SLOW CONVERGENCE FOR TEMPERED LANGEVIN DYNAMICS

The upper bounds in Theorem 3 show that, so long as the log-Sobolev constants remain well-behaved along the geometric tempering, Langevin with moving target can successfully sample from the target distribution. While this assumption holds in favorable cases where the proposal and target are both strongly log-concave, it may fail even when both distributions are well-conditioned yet not log-concave, as demonstrated in Theorem 4. Nevertheless, such poor functional inequalities do not necessarily rule out fast convergence of the geometric tempering because functional inequalities only govern mixing in the worst case, and so it is still possible *a priori* that our upper bounds are loose in such cases. The purpose of this section is to develop rigorous lower bounds for the geometric tempering in two simple examples where the log-Sobolev constants of the intermediate distributions are poor.

**Setup.** Throughout this section, we let the proposal distribution be the standard Gaussian $\nu := \mathcal{N}(0,1)$, as is common for the geometric tempering (Cabezas et al., 2023; Dai et al., 2020; Zhang et al., 2021; Thin et al., 2021). We work in the setting of a discrete temperature schedule but with continuous time inner Langevin iterations. In particular, we take as fixed a discrete temperature schedule $\lambda_0 = 0 < \lambda_1 < \cdots < \lambda_{K-1} < \lambda_K = 1$ and a sequence of times $T_1, \ldots, T_K$ that the inner Langevin iteration is run for. We then put $p_{T_0}^0 := \nu$ and define inductively $p_t^k$ to be the law at time $t$ of Langevin initialized at $p_{T_{k-1}}^{k-1}$ and run towards $\mu_k$. Our goal is then to study the convergence of the final output $p_{T_K}^K$ towards $\pi$, as a function of the temperature and time sequences, as well as the target $\pi$.

**Bi-modal target.** Since annealing is designed to overcome multi-modality and associated poor mixing in the target, we begin by studying a toy model of multi-modality. Specifically, we take a parameter $m > 0$, and consider

$$\nu := \mathcal{N}(0,1), \qquad \pi := \frac{1}{2}\mathcal{N}(0,1) + \frac{1}{2}\mathcal{N}(m,1). \tag{16}$$

It can be checked that $\pi$, as well as some of the intermediate distributions in the geometric tempering, have log-Sobolev constant exponential in the mode separation $m$. However, as we mentioned above, it remains possible *a priori* that the geometric tempering avoids this exponentially poor conditioning by following non-worst case distributions. The following result shows that this is not the case, and indeed that the total time spent on Langevin dynamics must be exponential in $m$ to ensure convergence.

**Theorem 8** *Suppose $\nu, \pi$ are as in Eq. 16 for $m \geq 11$. Then*

$$\mathrm{TV}(p_{T_K}^K, \pi) \geq \frac{1}{20} - 16 \cdot e^{-m^2/64} \cdot \sum_{k=1}^{K} T_k.$$

The proof of Theorem 8 can be found in Appendix D.2. Several remarks are in order. First, we emphasize that this result has no dependence whatsoever on the tempering schedule. Second, because of classical inequalities between divergences (Polyanskiy and Wu, 2022, Section 7.6), the same result also holds with $\sqrt{\frac{1}{2} \mathrm{KL}(\rho_T, \mu_k)}, \sqrt{\frac{1}{2} \chi^2(\rho_T, \mu_k)}$ on the left-hand side. And finally, we remark that other works have studied related tempering algorithms with a uniform, rather than Gaussian, proposal, and shown that in this case tempered dynamics can successfully sample from mixtures of Gaussians, so long as each mixture component has the same variance (Woodard et al., 2009b; Ge et al., 2018). Theorem 8 shows that, in contrast, tempered Langevin dynamics with Gaussian proposal can fail for mixtures of Gaussians with identical variance, and therefore additionally suggests an unfavorable property of Gaussian proposals as compared with uniform.

**Well-conditioned target.** The previous result establishes a simple case where tempering is a natural approach, yet the geometric tempering suffers from exponentially poor performance. Since the target distribution in that case has poor functional inequalities, a simpler sanity check for the geometric tempering is to establish its convergence when the target is actually well-conditioned. We now analyze an example similar to that of Theorem 4, where the target has decent log-Sobolev inequalities, and show that, nonetheless, the geometric tempering suffers from exponentially poor convergence, at least until the end of the tempering scheme. For some $m > 0$, put

$$\nu := \mathcal{N}(0, 1), \qquad \pi := \frac{1}{2}\mathcal{N}(m, 1) + \frac{1}{2}u_m, \tag{17}$$

where, as in section 4.1, $u_m$ is the probability distribution with density proportional to $e^{-\frac{1}{2}d^2(x, I_m)}$ for $I_m = [-m, 2m]$. Similar to Theorem 4, although $\pi$ has a reasonable log-Sobolev constant, the geometric tempering is partially bimodal. The next result shows that the geometric tempering suffers from exponentially slow convergence, at least until the tempering is almost at 1.

**Theorem 9** *Suppose $\nu, \pi$ are as in Eq. 17 for $m \geq 4$. Then $C_{LS}(\pi) \leq 324m^3$, yet for all $k \in [K]$*

$$\mathrm{TV}(p_{T_k}^k, \pi) \geq \frac{1}{5} - \delta_k - 10m \cdot \sqrt{\delta_k} \cdot \sum_{i=1}^{k} T_i,$$

*for $\delta_k := 6m^3 e^{-(1-\lambda_k)m^2/10}$.*

The proof of Theorem 9 can be found in Appendix D.3. This result demonstrates a simple case where geometric tempering suffers from exponentially slower convergence than Langevin dynamics, and to the best of our knowledge is the first result of this kind for the geometric tempering in the literature. As for Theorem 8, the statement holds with $\sqrt{\frac{1}{2} \mathrm{KL}(\rho_T, \mu_k)}, \sqrt{\frac{1}{2} \chi^2(\rho_T, \mu_k)}$ on the left-hand side. We finally remark that Theorems 8 and 9 are both consequences of a more general TV lower bound, Theorem 20 in Appendix D.1, which may be of further interest.

## 6 CONCLUSION

We provided the first convergence analysis of tempered Langevin dynamics, Eq. 9, in the literature, both for continuous (Theorem 1), and discrete (Theorem 3), time. Our bounds are naturally driven by the log-Sobolev constants of the geometric tempering path, and we identified in Theorem 4 a surprising degeneracy of the geometric tempering where it can make these constants exponentially worse

than that of the target. Restricting to the strongly log-concave setting, we rigorously established the optimal tempering schedule for our upper bounds in Proposition 6, and identified a regime where it is strictly distinct from vanilla Langevin. Finally, we developed rigorous lower bounds proving exponentially slow convergence for a bimodal target in Theorem 8, and even demonstrated a novel failure of the geometric tempering for a uni-modal target in Theorem 9, where it has exponentially worse performance than vanilla Langevin. Interesting questions for future work include developing a more complete understanding of the log-Sobolev constants along the geometric tempering, particularly for the uniform proposal, as well as identifying alternative paths for Langevin which have more favorable properties. In this connection, we mention some recent works which modify the geometric path so that mode weights are kept constant along the path (Bhatnagar and Randall, 2015; Tawn et al., 2018); it would be interesting to extend our analysis to these algorithms.

**Acknowledgements** A. Vacher, O. Chehab, and A. Korba were supported by funding from the French ANR JCJC WOS granted to A. Korba.

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

APPENDIX

The paper and appendix are organized as follows.

# A    CONVERGENCE OF CONTINUOUS-TIME TEMPERED LANGEVIN DYNAMICS

In this section, we give the proof of our continuous-time upper bound, Theorem 1. In Appendix A.1 we discuss the well-posedness of the continuous time dynamics. The proof of Theorem 1 is then given in Appendix A.2. In the next section, Appendix A.3, we derive from Theorem 1 guarantees for convergence to a given precision. In Appendix A.4 we give the proofs of the lemmas we used in the proof of Theorem 1.

## A.1    WELL-POSEDNESS OF CONTINUOUS TEMPERED LANGEVIN DYNAMICS

Notice that in Equation (9), the drift $b(t,x) = \nabla \log \mu_t(x)$ is time-dependent and the diffusion coefficient is constant. Still, existence and uniqueness of solutions can be guaranteed under mild assumptions. For instance, if $\nabla \log \nu$ and $\nabla \log \pi$ are $L_\nu$ and $L_\pi$-Lipschitz respectively, the drift is also Lipschitz with bounded constant $L \leq \max(L_\nu, L_\pi)$ (since the dynamic $\lambda_t$ is bounded between 0 and 1), hence satisfy the usual linear growth condition in the second argument. In this case, there exists a unique, continuous strong solution $(X_t)_{t \geq 0}$ adapted to the filtration generated by the Brownian motion which satisfies Equation (9) (Itô, 1951; Kotz et al., 2004). Strong uniqueness also means that if two processes satisfy this equation with the same initial conditions, their trajectories are almost surely indistinguishable. The law $(p_t)_{t \geq 0}$ of Equation (9) satisfies a Fokker-Planck equation (Bogachev et al., 2022) that can be written

$$\frac{\partial p_t}{\partial t} = \boldsymbol{\nabla} \cdot \left( p_t \nabla \log \left( \frac{p_t}{\mu_t} \right) \right) \ . \tag{18}$$

Generally, proving the existence and uniqueness of solutions for FPEs is more difficult than for SDEs, but one can also get under mild assumptions existence and uniqueness for solutions of Equation (18), under the same Lipschitzness assumptions on the score of proposal and target distributions, and using that the tempering dynamics are bounded in $[0, 1]$ (Bris and Lions, 2008, Proposition 2).

## A.2    PROOF OF THEOREM 1

In this section, we will prove Theorem 1 on the convergence rate of tempered Langevin dynamics. The proof is in two steps: first, we compute how the process $p_t$ tracks the moving target $\mu_t$, then we compute how well the moving target $\mu_t$ tracks the final target $\pi$.

**Step 1: contraction of** $\mathrm{KL}(p_t, \mu_t)$ **over time.**    Using the chain rule, we have

$$\frac{d}{dt} \mathrm{KL}(p_t, \mu_t) = \int \log \left( \frac{p_t}{\mu_t} \right) \frac{\partial p_t}{\partial t} + \int \frac{-p_t}{\mu_t} \frac{\partial \mu_t}{\partial t} := a + b.$$

The first term $a$ involves the Langevin dynamics given by the Fokker-Planck equation Eq. 18. Hence, using integration by parts and the inverse log-Sobolev constants defined in Eq. 8, we obtain:

$$a = \int \log \left( \frac{p_t}{\mu_t} \right) \boldsymbol{\nabla} \cdot \left( p_t \nabla \log \left( \frac{p_t}{\mu_t} \right) \right) = - \int p_t \left\| \nabla \log \left( \frac{p_t}{\mu_t} \right) \right\|^2$$

$$= - \mathrm{FD}(p_t, \mu_t) \leq -2\alpha_t \mathrm{KL}(p_t, \mu_t). \tag{19}$$

The second term $b$ is specific to the tempering scheme: it involves the tempering dynamics $\dot{\mu}_t$, which is zero when there is no tempering and are here determined by the tempering rule $\mu_t = c_{\lambda_t} \nu^{1-\lambda_t} \pi^{\lambda_t}$, where $c_{\lambda_t} = 1 / \int \nu^{1-\lambda_t} \pi^{\lambda_t}$ is a normalizing factor, that we will denote $c_t$ in this proof to alleviate notation. We can compute the (log) tempering dynamics

$$\frac{\partial \log \mu_t}{\partial t} = \frac{\partial}{\partial t} \left( \lambda_t \log \frac{\pi}{\nu} + \log \nu - \log \int \nu^{1-\lambda_t} \pi^{\lambda_t} \right) = \dot{\lambda}_t \log \frac{\pi}{\nu} - \frac{\frac{\partial}{\partial t} \int \nu^{1-\lambda_t} \pi^{\lambda_t}}{\int \nu^{1-\lambda_t} \pi^{\lambda_t}}$$

$$= \dot{\lambda}_t \log \frac{\pi}{\nu} - \dot{\lambda}_t \int \log \frac{\pi}{\nu} \frac{\nu^{1-\lambda_t} \pi^{\lambda_t}}{\int \nu^{1-\lambda_t} \pi^{\lambda_t}} = \dot{\lambda}_t \left( \log \frac{\pi}{\nu} - \mathbb{E}_{\mu_t} \left[ \log \frac{\pi}{\nu} \right] \right)$$

so that the second term becomes

$$b = \int p_t \frac{-\partial \log \mu_t}{\partial t} = \dot{\lambda}_t \left( \mathbb{E}_{\mu_t} \left[ \log \frac{\pi}{\nu} \right] - \mathbb{E}_{p_t} \left[ \log \frac{\pi}{\nu} \right] \right) \ .$$

To control this term we prove the following Lemma in Appendix A.4.

**Lemma 10** (Discrepancy between the laws of the sampling process and tempering path) *We have*

$$\mathbb{E}_{\mu_t}\Big[\log\frac{\pi}{\nu}\Big] - \mathbb{E}_{p_t}\Big[\log\frac{\pi}{\nu}\Big] \leq 2(L_\pi + L_\nu)\Big(\frac{2(d + b_\nu + b_\pi)}{a_\nu \wedge a_\pi} + \mathbb{E}_{p_0}[\|x\|^2]\Big).$$

Let $A$ be the right-hand side in Lemma 10, namely

$$A := 2(L_\pi + L_\nu)\Big(\frac{2(d + b_\nu + b_\pi)}{a_\nu \wedge a_\pi} + \mathbb{E}_{p_0}[\|x\|^2]\Big). \tag{20}$$

Then we obtain

$$b \leq A\dot{\lambda}_t. \tag{21}$$

Combining Eq. 19 and Eq. 21 we obtain

$$\frac{d}{dt}\,\mathrm{KL}(p_t, \mu_t) \leq -2\alpha_t\,\mathrm{KL}(p_t, \mu_t) + A\dot{\lambda}_t.$$

To conclude this step, we apply Grönwall's lemma (Mischler, 2019, Lemma 1.1) to yield

$$\mathrm{KL}(p_t, \mu_t) \leq \exp\Big(\int_0^t -2\alpha_s ds\Big)\,\mathrm{KL}(p_0, \mu_0) + A\int_0^t \dot{\lambda}_s \exp\Big(\int_s^t -2\alpha_v dv\Big)ds.$$

**Step 2: compare $\mathrm{KL}(p_t, \mu_t)$ and $\mathrm{KL}(p_t, \pi)$.**   This measures how well the moving target $\mu_t$ tracks the final target $\pi$. We write

$$\mathrm{KL}(p_t, \pi) = \mathrm{KL}(p_t, \mu_t) + \mathrm{KL}(p_t, \pi) - \mathrm{KL}(p_t, \mu_t)$$
$$= \mathrm{KL}(p_t, \mu_t) + \mathbb{E}_{p_t}\Big[\log\frac{\mu_t}{\pi}\Big]$$
$$= \mathrm{KL}(p_t, \mu_t) + (1 - \lambda_t)\mathbb{E}_{p_t}\Big[\log\Big(\frac{\nu}{\pi}\Big)\Big] + \log c_t.$$

We make the following observation on the log normalizing constants, proved in Appendix A.4.

**Lemma 11** (Bounding the normalizing constant of the law of the tempering path) *We have*

$$0 \leq \log c_t \leq \min\Big(\lambda_t\,\mathrm{KL}(\nu, \pi), (1 - \lambda_t)\,\mathrm{KL}(\pi, \nu)\Big)\ .$$

Using Lemma 11 to control the normalizing constant $c_t$, we obtain

$$\mathrm{KL}(p_t, \pi) \leq \mathrm{KL}(p_t, \mu_t)\ + (1 - \lambda_t)\Big(\mathbb{E}_{p_t}\Big[\log\Big(\frac{\nu}{\pi}\Big)\Big] + \mathbb{E}_\pi\Big[\log\Big(\frac{\pi}{\nu}\Big)\Big]\Big).$$

Finally, we control this last term using similar estimates as in Lemma 10; the proof is again deferred to Appendix A.4.

**Lemma 12** (Discrepancy between the laws of the sampling process and target) *We have*

$$\mathbb{E}_\pi\Big[\log\frac{\pi}{\nu}\Big] - \mathbb{E}_{p_t}\Big[\log\frac{\pi}{\nu}\Big] \leq 2(L_\pi + L_\nu)\Big(\frac{2(d + b_\nu + b_\pi)}{a_\nu \wedge a_\pi} + \mathbb{E}_{p_0}[\|x\|^2]\Big).$$

Applying this result we obtain

$$\mathrm{KL}(p_t, \pi) \leq \mathrm{KL}(p_t, \mu_t) + (1 - \lambda_t)A.$$

**Step 3: putting it together.**   We find

$$\mathrm{KL}(p_t, \pi) \leq (u_1) + (u_2) + (u_3) \tag{22}$$
$$(u_1) = \exp\Big(-2\int_0^t \alpha_s ds\Big)\mathrm{KL}(p_0, \mu_0)$$
$$(u_2) = (1 - \lambda_t)A$$
$$(u_3) = A\int_0^t \dot{\lambda}_s \exp\Big(-2\int_s^t \alpha_v dv\Big)ds\ .$$

A.3 CONTINUOUS-TIME CONVERGENCE IN PRECISION FORM

As in Vempala and Wibisono (2019), we can derive from Theorem 1 sufficient conditions for convergence to precision $\epsilon$ by setting each of the three terms in the upper-bound to be less than $\epsilon/3$.

**Corollary 13** *Suppose the assumptions of Theorem 1 hold. To achieve convergence with precision $\epsilon > 0$ such that $\mathrm{KL}(p_t, \pi) < \epsilon$, sufficient conditions on the time $t$ and schedule $\lambda(\cdot)$ are*

$$t > \max\left( \frac{1}{2\alpha_{\min}} \log\left( \frac{3\,\mathrm{KL}(p_0, \mu_0)}{\epsilon} \right), \frac{1}{\alpha_{\min}} \log\left( \frac{6A}{\epsilon} \right) \right) \tag{23}$$

$$\lambda_t > 1 - \frac{\epsilon}{3A} \tag{24}$$

$$\lambda_t < \frac{\epsilon}{6} + \lambda_{t/2}, \tag{25}$$

*where $\alpha_{\min} = \min_{s>0} \alpha_s$ is the smallest log-Sobolev constant among the path of tempered distributions.*

*Proof of Corollary 13.* We upper-bound the first term by $(u_1) \leq e^{-2t\alpha_{\min}} \mathrm{KL}(p_0, \mu_0)$ so that a sufficient condition for $(u_1) < \epsilon/3$ is $t > \frac{1}{2\alpha_{\min}} \log\left( \frac{3\,\mathrm{KL}(p_0, \mu_0)}{\epsilon} \right)$.

A sufficient condition $(u_2) < \epsilon/3$ is $\lambda_t \in ]1 - \frac{\epsilon}{3A}, 1]$. If $\lambda$ is bijective, this yields $t > \lambda^{-1}\left( 1 - \frac{\epsilon}{3A} \right)$.

Finally, we upper-bound the third term:

$$(u_3) = A \int_0^t \dot{\lambda}_s \exp\left( -2 \int_s^t \alpha_v dv \right) ds \tag{26}$$

$$= A\left( \int_0^{t/2} \dot{\lambda}_s \exp\left( -2 \int_s^t \alpha_v dv \right) ds + \int_{t/2}^t \dot{\lambda}_s \exp\left( -2 \int_s^t \alpha_v dv \right) ds \right) \tag{27}$$

$$\leq A\left( \exp\left( -2 \int_{t/2}^t \alpha_v dv \right) \int_0^{t/2} \dot{\lambda}_s + \int_{t/2}^t \dot{\lambda}_s ds \right) \tag{28}$$

$$\leq A\left( \exp(-t\alpha_{\min})(\lambda_{t/2} - \lambda_0) + (\lambda_t - \lambda_{t/2}) \right) \tag{29}$$

$$\leq A\left( \exp(-t\alpha_{\min}) + (\lambda_t - \lambda_{t/2}) \right) \tag{30}$$

so that sufficient conditions for $(u_3) < \frac{\epsilon}{3}$ can be obtained by setting each of the two terms smaller than $\frac{\epsilon}{6}$, yielding $t > \frac{1}{\alpha_{\min}} \log \frac{6A}{\epsilon}$ and $\lambda_t - \lambda_{t/2} < \epsilon/6$. $\qquad \square$

A.4 PROOFS OF TECHNICAL LEMMAS FROM APPENDIX A.2

In this section, we give the deferred proofs from Appendix A.2. We first state and prove two observations that will be useful, before turning to the deferred proofs. The following proposition translates the dissipativity assumptions into a condition on the tails of the proposal and target.

**Proposition 14** (Tails of the proposal and target) *For all $x \in \mathbb{R}^d$, we have*

$$V_\nu(x) - \inf_{y \in \mathbb{R}^d} V_\nu(y) \leq 2L_\nu\left( \|x\|^2 + \frac{b_\nu}{a_\nu} \right), \qquad V_\pi(x) - \inf_{y \in \mathbb{R}^d} V_\pi(y) \leq 2L_\pi\left( \|x\|^2 + \frac{b_\pi}{a_\pi} \right).$$

*Proof of Proposition 14.* Observe that because

$$\langle \nabla V_\nu(x), x \rangle \geq a_\nu \|x\|^2 - b_\nu,$$

the function $V_\nu$ must be minimized by some point $x_0 \in B(0, \sqrt{\frac{b_\nu}{a_\nu}})$. Therefore, for all $x \in \mathbb{R}^d$,

$$V_\nu(x) - \inf_{y \in \mathbb{R}^d} V_\nu(y) = V_\nu(x) - V_\nu(x_0) \leq L_\nu \|x - x_0\|^2 \leq 2L_\nu\left( \|x\|^2 + \frac{b_\nu}{a_\nu} \right).$$

The proof for $V_\pi$ is identical, so we omit it. $\qquad \square$

The next observation is our main quantitative use of the dissipativity assumption, and crucial for our control on the terms arising from the tempering dynamics in the continuous time convergence proof.

**Lemma 15** (Bounded second moment along the dynamics) *For all $t \geq 0$,*

$$\mathbb{E}_{p_t}[\|x\|^2] \leq \max\left(\mathbb{E}_{p_0}[\|x\|^2], \frac{d + b_\nu + b_\pi}{a_\nu \wedge a_\pi}\right).$$

*Proof of Lemma 15.* Using the Fokker-Planck equation Eq. 18 and integration by parts several times, we find

$$\partial_t \mathbb{E}_{p_t}[\|x\|^2] = \int \|x\|^2 \nabla \cdot \left(p_t \nabla \log \frac{p_t}{\mu_t}\right)$$

$$= -2 \int \langle x, \nabla \log p_t - \nabla \log \mu_t \rangle p_t$$

$$= 2d - 2 \int \langle x, (1 - \lambda_t)\nabla V_\nu + \lambda_t \nabla V_\pi \rangle p_t \quad .$$

Applying the dissipativity assumption, we obtain

$$\partial_t \mathbb{E}_{p_t}[\|x\|^2] \leq 2(d + (1 - \lambda_t)b_\nu + \lambda_t b_\pi) - 2((1 - \lambda_t)a_\nu + \lambda_t a_\pi)\mathbb{E}_{p_t}[\|x\|^2]$$

$$\leq 2(d + b_\nu + b_\pi) - 2(a_\nu \wedge a_\pi)\mathbb{E}_{p_t}[\|x\|^2].$$

Therefore, as soon as $\mathbb{E}_{p_t}[\|x\|^2]$ exceeds the level $(d + b_\nu + b_\pi)/(a_\nu \wedge a_\pi)$, it must start decreasing. The result follows. □

We now give the proofs of the Lemmas from Appendix A.2.

*Proof of Lemma 10.* Use Proposition 14 to obtain

$$\mathbb{E}_{\mu_t}\left[\log \frac{\pi}{\nu}\right] - \mathbb{E}_{p_t}\left[\log \frac{\pi}{\nu}\right] = \mathbb{E}_{\mu_t}[V_\nu - V_\pi] + \mathbb{E}_{p_t}[V_\pi - V_\nu]$$

$$\leq \mathbb{E}_{\mu_t}[V_\nu - \inf_{y \in \mathbb{R}^d} V_\nu(y)] + \mathbb{E}_{p_t}[V_\pi - \inf_{y \in \mathbb{R}^d} V_\pi(y)]$$

$$\leq 2L_\nu \mathbb{E}_{\mu_t}[\|x\|^2] + 2L_\pi \mathbb{E}_{p_t}[\|x\|^2] + 2L_\nu \frac{b_\nu}{a_\nu} + 2L_\pi \frac{b_\pi}{a_\pi}.$$

Let $M$ be the quantity appearing on the right-hand side of Lemma 15. Then we obtain

$$\mathbb{E}_{\mu_t}\left[\log \frac{\pi}{\nu}\right] - \mathbb{E}_{p_t}\left[\log \frac{\pi}{\nu}\right] \leq 2L_\nu \mathbb{E}_{\mu_t}[\|x\|^2] + 2L_\pi\left(M + \frac{b_\pi}{a_\pi}\right) + 2L_\nu \frac{b_\nu}{a_\nu}.$$

To handle $\mathbb{E}_{\mu_t}[\|x\|^2]$, we apply the dissipativity assumption to yield

$$\mathbb{E}_{\mu_t}[\|x\|^2] \leq \frac{1}{(1 - \lambda_t)a_\nu + \lambda_t a_\pi} \mathbb{E}_{\mu_t}\left[\langle(1 - \lambda_t)\nabla V_\nu + \lambda_t V_\pi, x\rangle + (1 - \lambda_t)b_\nu + \lambda_t b_\pi\right].$$

Integration by parts gives

$$\mathbb{E}_{\mu_t}[\|x\|^2] \leq \frac{d + (1 - \lambda_t)b_\nu + \lambda_t b_\pi}{(1 - \lambda_t)a_\nu + \lambda_t a_\pi} \leq \frac{d + b_\nu + b_\pi}{a_\nu \wedge a_\pi}.$$

Hence,

$$\mathbb{E}_{\mu_t}\left[\log \frac{\pi}{\nu}\right] - \mathbb{E}_{p_t}\left[\log \frac{\pi}{\nu}\right] \leq 2L_\pi\left(\frac{b_\pi}{a_\pi} + \max\left(\mathbb{E}_{p_0}[\|x\|^2], \frac{d + b_\nu + b_\pi}{a_\nu \wedge a_\pi}\right)\right)$$

$$+ 2L_\nu\left(\frac{b_\nu}{a_\nu} + \frac{d + b_\nu + b_\pi}{a_\nu \wedge a_\pi}\right).$$

Using $\max(x, y) \leq x + y$ when $x, y \geq 0$ and

$$\frac{b_\nu}{a_\nu}, \frac{b_\pi}{a_\pi} \leq \frac{d + b_\nu + b_\pi}{a_\nu \wedge a_\pi},$$

we conclude. □

*Proof of Lemma 11.* We have

$$\log c_t = \log \left( \frac{1}{\int \nu^{1-\lambda_t} \pi^{\lambda_t}} \right) = -\log \int \nu^{1-\lambda_t} \pi^{\lambda_t}$$

The lower bound is immediate from Hölder's inequality. For the upper bound, write

$$\log c_t = -\log \mathbb{E}_\nu \left[ \left( \frac{\pi}{\nu} \right)^{\lambda_t} \right] \leq \lambda_t \mathbb{E}_\nu \left[ \log \frac{\nu}{\pi} \right] = \lambda_t \operatorname{KL}(\nu, \pi)$$

using Jensen's inequality. Similarly, we can write

$$\log c_t = -\log \mathbb{E}_\pi \left[ \left( \frac{\nu}{\pi} \right)^{1-\lambda_t} \right] \leq (1-\lambda_t) \mathbb{E}_\pi \left[ \log \frac{\pi}{\nu} \right] = (1-\lambda_t) \operatorname{KL}(\pi, \nu) \ ,$$

and therefore conclude that

$$\log c_t \leq \min \left( \lambda_t \operatorname{KL}(\nu, \pi), (1-\lambda_t) \operatorname{KL}(\pi, \nu) \right) \ . \qquad \square$$

*Proof of Lemma 12.* Denoting by $V_\pi^*$ (resp. $V_\nu^*$) the infimum of $V_\pi$ (resp. $V_\nu$), we have

$$\mathbb{E}_\pi \left[ \log \frac{\pi}{\nu} \right] - \mathbb{E}_{p_t} \left[ \log \frac{\pi}{\nu} \right] = \int p_t (V_\pi - V_\pi^* - (V_\nu - V_\nu^*)) + \int d\pi (V_\nu - V_\nu^*) - (V_\pi - V_\pi^*).$$

Using Proposition 14, we obtain the upper-bound

$$\mathbb{E}_\pi \left[ \log \frac{\pi}{\nu} \right] - \mathbb{E}_{p_t} \left[ \log \frac{\pi}{\nu} \right] \leq 2L_\pi \left( \mathbb{E}_{p_t}[\|x\|^2] + \frac{b_\pi}{a_\pi} \right) + 2L_\nu \left( \mathbb{E}_\pi[\|x\|^2] + \frac{b_\nu}{a_\nu} \right).$$

Let $M$ be the quantity appearing on the right-hand side of Lemma 15. Then

$$\mathbb{E}_\pi \left[ \log \frac{\pi}{\nu} \right] - \mathbb{E}_{p_t} \left[ \log \frac{\pi}{\nu} \right] \leq 2L_\pi \left( M + \frac{b_\pi}{a_\pi} \right) + 2L_\nu \left( \mathbb{E}_\pi[\|x\|^2] + \frac{b_\nu}{a_\nu} \right).$$

On the other hand, a direct application of the dissipativity condition Assumption 2 implies

$$\mathbb{E}_\pi[\|x\|^2] \leq \frac{1}{a_\pi} \mathbb{E}_\pi[\langle x, \nabla V_\pi(x) \rangle + b_\pi] = \frac{d + b_\pi}{a_\pi} \leq \frac{d + b_\pi + b_\nu}{a_\nu \wedge a_\pi}.$$

Therefore

$$\mathbb{E}_\pi \left[ \log \frac{\pi}{\nu} \right] - \mathbb{E}_{p_t} \left[ \log \frac{\pi}{\nu} \right] \leq 2L_\pi \left( \max \left( \mathbb{E}_{p_0}[\|x\|^2], \frac{d + b_\nu + b_\pi}{a_\nu \wedge a_\pi} \right) + \frac{b_\pi}{a_\pi} \right.$$
$$\left. + 2L_\nu \frac{d + b_\pi + b_\nu}{a_\nu \wedge a_\pi} + 2L_\nu \frac{b_\nu}{a_\nu} \right).$$

Again, bounding $\max(x, y) \leq x + y$ for $x, y \geq 0$ and using

$$\frac{b_\nu}{a_\nu}, \ \frac{b_\pi}{a_\pi} \leq \frac{d + b_\nu + b_\pi}{a_\nu \wedge a_\pi},$$

we conclude. $\qquad \square$

## B  CONVERGENCE OF DISCRETE-TIME TEMPERED LANGEVIN DYNAMICS

In this section we give proof of our discrete-time upper bound, Theorem 3. In Appendix B.1 we outline the proof of Theorem 3. In Appendix B.2, we derive from Theorem 3 sufficient conditions to precision $\operatorname{KL}(p_k, \pi) < \epsilon$. And in Appendix B.3 we give the deferred proof from Appendix B.1.

### B.1  PROOF OF THEOREM 3

We next prove Theorem 3. Broadly speaking, the proof is in two steps: first, we compute how the process $p_k$ tracks the moving target $\mu_k$, then we compute how well the moving target $\mu_k$ tracks the final target $\pi$.

**Step 1: contraction of** $\mathrm{KL}(p_k, \mu_k)$**.** Using Vempala and Wibisono (2019, Lemma 3), under Assumptions 1 and 2 and $h_k \leq \frac{\alpha_k}{4L_k^2}$, we obtain

$$\mathrm{KL}(p_k, \mu_k) \leq \mathrm{KL}(p_{k-1}, \mu_k)e^{-\alpha_k h_k} + 6h_k^2 dL_k^2 \ . \tag{31}$$

Next, we can compute the (log) tempering dynamics of $\mu_k$

$$\log \mu_{k-1} - \log \mu_k = (\lambda_k - \lambda_{k-1})\log\frac{\nu}{\pi} - \log\frac{\int \nu^{1-\lambda_{k-1}}\pi^{\lambda_{k-1}}}{\int \nu^{1-\lambda_k}\pi^{\lambda_k}}$$

$$= (\lambda_k - \lambda_{k-1})\log\frac{\nu}{\pi} - \log\int \frac{\nu^{1-\lambda_{k-1}}\pi^{\lambda_{k-1}}}{\nu^{1-\lambda_k}\pi^{\lambda_k}}\frac{\nu^{1-\lambda_k}\pi^{\lambda_k}}{\int \nu^{1-\lambda_k}\pi^{\lambda_k}}$$

$$= (\lambda_k - \lambda_{k-1})\log\frac{\nu}{\pi} - \log\mathbb{E}_{\mu_k}\left[\left(\frac{\nu}{\pi}\right)^{\lambda_k - \lambda_{k-1}}\right]$$

$$\leq (\lambda_k - \lambda_{k-1})\left(\log\frac{\nu}{\pi} - \mathbb{E}_{\mu_k}\left[\log\frac{\nu}{\pi}\right]\right),$$

where we used Jensen's inequality at the last step. Thus

$$\mathrm{KL}(p_{k-1}, \mu_k) = \mathrm{KL}(p_{k-1}, \mu_{k-1}) + \mathrm{KL}(p_{k-1}, \mu_k) - \mathrm{KL}(p_{k-1}, \mu_{k-1})$$

$$= \mathrm{KL}(p_{k-1}, \mu_{k-1}) + \int p_{k-1}\log\frac{\mu_{k-1}}{\mu_k}$$

$$\leq \mathrm{KL}(p_{k-1}, \mu_{k-1}) + (\lambda_k - \lambda_{k-1})\left(\mathbb{E}_{\mu_k}\left[\log\frac{\pi}{\nu}\right] - \mathbb{E}_{p_{k-1}}\left[\log\frac{\pi}{\nu}\right]\right).$$

Using Proposition 14 we can upper bound the difference of expectations via

$$\mathbb{E}_{\mu_k}\left[\log\frac{\pi}{\nu}\right] - \mathbb{E}_{p_{k-1}}\left[\log\frac{\pi}{\nu}\right] \leq 2L_\nu\left(\mathbb{E}_{\mu_k}[\|x\|^2] + \frac{b_\nu}{a_\nu}\right) + 2L_\pi\left(\mathbb{E}_{p_{k-1}}[\|x\|^2] + \frac{b_\pi}{a_\pi}\right)$$

The next Lemma, proved in Appendix B.3 controls the second moment of $p_k$.

**Lemma 16** (Bounded second moment along the dynamics) *Suppose* $h_k \leq \min\left(1, \frac{a_\pi \wedge a_\nu}{2(L_\pi + L_\nu)^2}\right)$. *Then*

$$\mathbb{E}_{p_k}[\|x\|^2] \leq \max\left(\mathbb{E}_{p_0}[\|x\|^2], \frac{2(3(b_\pi + b_\nu)/2 + d)}{a_\pi \wedge a_\mu \wedge 1}\right) \ .$$

For the second moment of $\mu_k$ we use the following bound from the proof of Lemma 10

$$\mathbb{E}_{\mu_k}[\|x\|^2] \leq \frac{d + b_\nu + b_\pi}{a_\nu \wedge a_\pi}.$$

To combine these bounds, let us define

$$A' := 2(L_\pi + L_\nu)\left\{\max\left(\mathbb{E}_{p_0}[\|x\|^2], \frac{2(3(b_\pi + b_\nu)/2 + d)}{a_\pi \wedge a_\nu \wedge 1}\right) + \frac{3(d + b_\nu + b_\pi)}{a_\pi \wedge a_\nu}\right\}. \tag{32}$$

Plugging into Eq. 31, we obtain

$$\mathrm{KL}(p_k, \mu_k) \leq \mathrm{KL}(p_{k-1}, \mu_{k-1})e^{-\alpha_k h_k} + (\lambda_k - \lambda_{k-1})A'e^{-\alpha_k h_k} + 6h_k^2 dL_k^2.$$

Now put $\psi_i^k := \sum_{j=i}^k \alpha_j h_j$. Unrolling the recursion, we obtain

$$\mathrm{KL}(p_k, \mu_k) \leq e^{-\psi_1^k}\mathrm{KL}(p_0, \mu_0) + \sum_{i=1}^k \left(A'(\lambda_i - \lambda_{i-1})e^{-\alpha_i h_i} + 6h_i^2 dL_i^2\right)e^{-\psi_{i+1}^k}. \tag{33}$$

**Step 2: compare** $\mathrm{KL}(p_k, \mu_k)$ **and** $\mathrm{KL}(p_k, \pi)$**.** Here we use Lemma 11 to control the log-normalizing constants and obtain

$$\mathrm{KL}(p_k, \pi) = \mathrm{KL}(p_k, \mu_k) + \mathrm{KL}(p_k, \pi) - \mathrm{KL}(p_k, \mu_k)$$

$$= \mathrm{KL}(p_k, \mu_k) + \mathbb{E}_{p_k}\left[\log\frac{\mu_k}{\pi}\right]$$

$$= \mathrm{KL}(p_k, \mu_k) + (1 - \lambda_k)\mathbb{E}_{p_k}\left[\log\frac{\nu}{\pi}\right] + \log c_k$$

$$\leq \mathrm{KL}(p_k, \mu_k) + (1 - \lambda_k)\left(\mathbb{E}_{p_k}\left[\log\frac{\nu}{\pi}\right] + \mathrm{KL}(\pi, \nu)\right).$$

To control the term in parentheses, we apply Proposition 14 to yield

$$\mathbb{E}_{p_k}\left[\log\frac{\nu}{\pi}\right] + \mathrm{KL}(\pi,\nu) \le 2L_\pi\left(\mathbb{E}_{p_k}[\|x\|^2] + \frac{b_\pi}{a_\pi}\right) + 2L_\nu\left(\mathbb{E}_\pi[\|x\|^2] + \frac{b_\nu}{a_\nu}\right).$$

Using Lemma 15 and the following bound from the proof of Lemma 12

$$\mathbb{E}_\pi[\|x\|^2] \le \frac{d + b_\pi + b_\nu}{a_\nu \wedge a_\pi},$$

we find

$$\mathbb{E}_{p_k}\left[\log\frac{\nu}{\pi}\right] + \mathrm{KL}(\pi,\nu) \le A',$$

for $A'$ as in Eq. 32.

**Step 3: putting it together.** Combining the results of steps 1 and 2, we can finally write

$$\mathrm{KL}(p_t,\pi) \le (v_1) + (v_2) + (v_3) + (v_4) \tag{34}$$

$$(v_1) = \exp\left(-\sum_{j=1}^k \alpha_j h_j\right)\mathrm{KL}(p_0,\mu_1),$$

$$(v_2) = (1-\lambda_k)A',$$

$$(v_3) = A'\sum_{i=1}^k (\lambda_i - \lambda_{i-1})\exp\left(-\sum_{j=i}^k \alpha_j h_j\right),$$

$$(v_4) = 6\sum_{i=1}^k h_k^2 dL_k^2 \exp\left(-\sum_{j=i+1}^k \alpha_j h_j\right).$$

### B.2 DISCRETE-TIME CONVERGENCE IN PRECISION FORM

**Corollary 17** *To achieve convergence with precision $\epsilon > 0$ such that $\mathrm{KL}(p_k,\pi) < \epsilon$, sufficient conditions are*

$$h < \min\left(\frac{1}{4\alpha_{\min}}, \frac{\alpha_{\min}}{96L_{\max}^2 d}\epsilon, \frac{\alpha_{\min}}{4(L_\pi + L_\nu)^2}, \frac{a_\pi \wedge a_\nu}{2(L_\pi + L_\nu)^2}, 1\right) \tag{35}$$

$$k > \max\left(\frac{1}{h\alpha_{\min}}\log\left(\frac{4\,\mathrm{KL}(p_0,\mu_1)}{\epsilon}\right), \frac{2}{h\alpha_{\min}}\log\left(\frac{8A'}{\epsilon}\right)\right) \tag{36}$$

$$\lambda_k > 1 - \frac{\epsilon}{4A'} \tag{37}$$

$$\lambda_k - \lambda_{\lfloor k/2\rfloor} < \frac{\epsilon}{24A'} \tag{38}$$

*where $\alpha_{\min} = \min_{s>0}\alpha_s$ and $L_{\max} = \max_{s>0}L_s$ denote the smallest (resp. largest) log-Sobolev (resp. smoothness) constant among the path of tempered distributions.*

*Proof of Corollary 17.* Again, as in Vempala and Wibisono (2019), we can derive sufficient conditions for convergence with a given precision $\mathrm{KL}(p_k,\pi) < \epsilon$ by setting each of the four terms in the upper-bound inferior to $\epsilon/4$.

First, we upper-bound the fourth term:

$$(v_4) \le 6h^2 dL_{\max}^2 \sum_{i=1}^k e^{-(k-i)h\alpha_{\min}} = 6h^2 dL_{\max}^2 \frac{1-e^{-kh\alpha_{\min}}}{1-e^{-h\alpha_{\min}}} \tag{39}$$

$$\le 6h^2 dL_{\max}^2 \frac{1}{1-e^{-h\alpha_{\min}}} \le 6h^2 dL_{\max}^2 \frac{4}{3h\alpha_{\min}} = \frac{8hdL_{\max}^2}{\alpha_{\min}} \tag{40}$$

where in the last inequality, similarly to Vempala and Wibisono (2019), we use that $1 - e^{-c} \ge \frac{3}{4}c$ for $0 < c = h\alpha_{\min} \le \frac{1}{4}$ which holds assuming that $h \le \frac{1}{4\alpha_{\min}}$. Thus, a sufficient condition for $(v_4) \le \epsilon/4$ is $h \le \frac{\alpha_{\min}}{32L_{\max}^2 d}\epsilon$.

Next, we upper-bound the first term by $(v_1) \leq e^{-kh\alpha_{\min}} \mathrm{KL}(p_0, \mu_0)$ so that a sufficient condition for $(v_1) < \epsilon/4$ is $k > \frac{1}{h\alpha_{\min}} \log\left(\frac{4\,\mathrm{KL}(p_0,\mu_0)}{\epsilon}\right)$.

A sufficient condition $(v_2) < \epsilon/4$ is $\lambda_k \in ]1 - \frac{\epsilon}{4A'}, 1]$.

Finally, we upper-bound the third term:

$$(v_3) := A' \sum_{i=1}^{k} (\lambda_i - \lambda_{i-1}) \exp\left(-h \sum_{j=i}^{k} \alpha_j\right) \tag{41}$$

$$= A'\left(\sum_{i=1}^{\lfloor k/2 \rfloor} (\lambda_i - \lambda_{i-1}) \exp\left(-h \sum_{j=i}^{k} \alpha_j\right) + \sum_{i=\lfloor k/2 \rfloor}^{k} (\lambda_i - \lambda_{i-1}) \exp\left(-h \sum_{j=i}^{k} \alpha_j\right)\right) \tag{42}$$

$$\leq A'\left(\exp\left(-h \sum_{j=\lfloor k/2 \rfloor}^{k} \alpha_j\right) \sum_{i=1}^{\lfloor k/2 \rfloor} (\lambda_i - \lambda_{i-1}) + \sum_{i=\lfloor k/2 \rfloor}^{k} (\lambda_i - \lambda_{i-1})\right) \tag{43}$$

$$\leq A'\left(\exp(-hk\alpha_{\min}/2)(\lambda_{\lfloor k/2 \rfloor} - \lambda_1) + (\lambda_k - \lambda_{\lfloor k/2 \rfloor})\right) \tag{44}$$

$$\leq A' \exp(-hk\alpha_{\min}/2) + A'(\lambda_k - \lambda_{\lfloor k/2 \rfloor}) \tag{45}$$

so that sufficient conditions for $(v_3) < \frac{\epsilon}{4}$ can be obtained by setting each of the two terms smaller than $\frac{\epsilon}{8}$, yielding $k > \frac{2}{\alpha_{\min}h} \log \frac{8A'}{\epsilon}$ and $\lambda_k - \lambda_{\lfloor k/2 \rfloor} < \frac{\epsilon}{8A'}$. $\qquad\square$

### B.3 TECHNICAL LEMMA FROM APPENDIX B.1

In this section, we prove Lemma 16.

*Proof of Lemma 16.* We recall that the particles follow the recursion

$$X_{k+1} = X_k - h_k \nabla V_{\lambda_k}(X_k) + \sqrt{2h_k} z_k,$$

where $z_k$ follows a standard normal distribution on $\mathbb{R}^d$ and $X_k \sim p_k$. Hence, we have

$$\mathbb{E}[\|X_{k+1}\|^2] = \mathbb{E}[\|X_k - h_k \nabla V_{\lambda_k}(X_k) + \sqrt{2h_k} z_k\|^2] \tag{46}$$

$$= \mathbb{E}[\|X_k - h_k \nabla V_{\lambda_k}(X_k)\|^2] + 2dh_k \tag{47}$$

$$= \mathbb{E}[\|X_k\|^2] - 2h_k \mathbb{E}[X_k^\top \nabla V_{\lambda_k}(X_k)] + h_k^2 \mathbb{E}[\|\nabla V_{\lambda_k}(X_k)\|^2] + 2dh_k. \tag{48}$$

The dissipativity assumption yields $\mathbb{E}[X_k^\top \nabla V_{\lambda_k}(X_k)] \geq (a_\pi \wedge a_\nu)\mathbb{E}[\|X_k\|^2] - (b_\nu + b_\pi)$ and as in the proof of Proposition 14 we have $\|\nabla V_{\lambda_k}(X_k)\|^2 \leq 2(L_\pi + L_\nu)^2(\|X_k\|^2 + \frac{b_\nu + b_\pi}{a_\mu \wedge a_\pi})$ so that

$$\mathbb{E}[\|X_{k+1}\|^2] \leq \mathbb{E}[\|X_k\|^2] - 2h_k((a_\pi \wedge a_\nu)\mathbb{E}[\|X_k\|^2] - (b_\nu + b_\pi)) + 2dh_k$$
$$+ 2h_k^2(L_\pi + L_\nu)^2\left(\mathbb{E}[\|X_k\|^2] + \frac{b_\nu + b_\pi}{a_\mu \wedge a_\pi}\right).$$

Letting $L := L_\pi + L_\nu$, $a := a_\nu \wedge a_\pi$ and $b := b_\nu + b_\pi$ we obtain

$$\mathbb{E}[\|X_{k+1}\|^2] \leq \mathbb{E}[\|X_k\|^2] - 2h_k\left((a - h_k L^2)\mathbb{E}[\|X_k\|^2] - (b + d) - h_k \frac{Lb}{a}\right)$$
$$= \mathbb{E}[\|X_k\|^2](1 - 2h_k(a - h_k L^2)) + 2h_k\left(b + d + h_k \frac{L^2 b}{a}\right).$$

Now by induction, assume that $\mathbb{E}[\|X_k\|^2] \leq \max(\mathbb{E}_{p_0}[\|x\|^2], \frac{2(3b/2+d)}{a\wedge 1})$. By construction on $h_k$, $a - h_k L^2 > 0$ hence if $\mathbb{E}[\|X_k\|^2] > \frac{b+d+h_k L^2 b/a}{2h_k(a-h_k L^2)}$, then $\mathbb{E}[\|X_{k+1}\|^2] < \mathbb{E}[\|X_k\|^2]$ and in particular $\mathbb{E}[\|X_{k+1}\|^2] \leq \max(\mathbb{E}_{p_0}[\|x\|^2], \frac{2(3b/2+d)}{a\wedge 1})$. Now, if $\mathbb{E}[\|X_k\|^2] \leq \frac{b+d+h_k L^2 b/a}{2h_k(a-h_k L^2)}$, then if

$1 > 2h_k(a - h_k L^2)$ we have

$$\mathbb{E}[\|X_{k+1}\|^2] \leq \frac{b + d + h_k L^2 b/a}{a - h_k L^2}(1 - 2h_k(a - h_k L^2)) + 2h_k(b + d + h_k L^2 b/a) \tag{49}$$

$$= \frac{b + d + h_k L^2 b/a}{a - h_k L^2} \tag{50}$$

$$\leq \frac{2(3b/2 + d)}{a}, \tag{51}$$

by construction on $h_k$. Conversely, if $1 \leq 2h_k(a - h_k L^2)$, then

$$\mathbb{E}[\|X_{k+1}\|^2] \leq 2h_k\left(b + d + h_k \frac{L^2 b}{a}\right)$$
$$\leq 2(3b/2 + d).$$

$\square$

## C  OPTIMIZATION OF CONTINUOUS-TIME TEMPERED LANGEVIN DYNAMICS

### C.1  PROOF OF COROLLARY 5

We will rewrite and bound the terms $(u_1)$, $(u_2)$ and $(u_3)$ in the upper bound of Theorem 1 provided in Eq. 34, so that the role of the tempering schedule $\lambda(\cdot)$ is more explicit, at the expense of a looser upper-bound.

**Rewriting $(u_3)$.**  We begin with the third term, using integration by parts to write

$$(u_3) := A \int_0^t \dot{\lambda}_s \exp\left(-2 \int_s^t \alpha_v dv\right) ds$$
$$= A\left(\lambda_t - \lambda_0 \exp\left(-2 \int_0^t \alpha_v dv\right) - 2 \int_0^t \lambda_s \alpha_s \exp\left(-2 \int_s^t \alpha_v dv\right) ds\right) .$$

**Recalling $(u_2)$.**  Recall that

$$(u_2) = A(1 - \lambda_t) .$$

**Rewriting $(u_1)$.**  We recall that $(u_1) = \exp\left(-2 \int_0^t \alpha_v dv\right) \mathrm{KL}(p_0, \mu_0)$. Assuming that the process is initialized at the proposal distribution $p_0 = \nu$, we can use the tempering rule $\mu_t = c_t \nu^{1-\lambda_t} \pi^{\lambda_t}$ to rewrite the KL term

$$\mathrm{KL}(\nu, \mu_0) = \lambda_0 \mathrm{KL}(\nu, \pi) - \log c_0.$$

By positivity of $\log c_0$, we can write $-\log c_0 \leq \lambda_0 \mathrm{KL}(\pi, \nu)$. In particular

$$\mathrm{KL}(\nu, \mu_0) \leq \lambda_0(\mathrm{KL}(\nu, \pi) + \mathrm{KL}(\pi, \nu))$$
$$= \lambda_0\left(\int [V_\pi(x) - V_\pi^* - (V_\nu(x) - V_\nu^*)] d\nu(x) + \int [V_\nu(x) - V_\nu^* - (V_\pi(x) - V_\pi^*)] d\pi(x)\right)$$
$$\leq \lambda_0 A.$$

**Upper bounding the sum.**  We can now combine upper bounds on $(u_1)$, $(u_2)$ and $(u_3)$ and simplify the result, using that the terms $\lambda_0 A \exp\left(-2 \int_0^t \alpha_v dv\right)$ and $A\lambda_t$ cancel out, to finally yield

$$\mathrm{KL}(p_t, \pi) \leq A \cdot G_t(\lambda), \qquad G_t(\lambda) := 1 - 2 \int_0^t \lambda_s \alpha_s \exp\left(-2 \int_s^t \alpha_v dv\right) ds. \tag{52}$$

**Optimal schedule when** $\alpha_\pi \geq \alpha_\nu$    Now, let $\Phi_s := \exp\left(-2\int_s^t \alpha_u du\right)$. We have

$$G_t(\lambda) = 1 - 2\int_0^t \lambda_s \alpha_s \Phi_s ds.$$

Applying integration by parts, we find

$$G_t(\lambda) = 1 - [\lambda_s \Phi_s]_0^t + \int_0^t \dot{\lambda}_s \Phi_s ds = 1 - \lambda_t + \lambda_0 \Phi_0 + \int_0^t \dot{\lambda}_s \Phi_s ds.$$

Since $\dot{\lambda}_s \geq 0$ and $\Phi_s \geq e^{-2(t-s)\alpha_\pi}$ (using the assumption $\alpha_\pi \geq \alpha_\nu$), it follows that

$$G_t(\lambda) \geq 1 - \lambda_t + \lambda_0 e^{-2t\alpha_\pi} + \int_0^t \dot{\lambda}_s e^{-2(t-s)\alpha_\pi} ds.$$

Reversing the integration by parts, we obtain

$$G_t(\lambda) \geq 1 - 2\int_0^t \lambda_s \alpha_\pi e^{-2(t-s)\alpha_\pi} ds \geq 1 - 2\int_0^t \alpha_\pi e^{-2(t-s)\alpha_\pi} ds,$$

where the last inequality uses $\lambda_s \leq 1$. Recognizing the right-hand side as $G_t$ evaluated at schedule $\lambda \equiv 1$, we conclude.

## C.2    PROOF OF PROPOSITION 6

*Sketch of proof.*  We first make the change of variable $\Phi(s) = \exp\left(-\int_s^t \alpha_u du\right)$ so that the problem can be re-written as

$$\sup_{\Phi \in I_t} \frac{1}{\alpha_\nu - \alpha_\pi}\left(\frac{\alpha_\nu}{2} - \int_0^t \dot{\Phi}_s^2 ds - \frac{\alpha_\nu}{2}\Phi_0^2\right),$$

where $I_t$ is the set of strictly positive functions $\Phi : [0, t] \mapsto \mathbb{R}$ with weak second derivative and such that $\alpha_\pi \Phi \leq \dot{\Phi} \leq \alpha_\nu \Phi$ and $\Phi \ddot{\Phi} \leq \dot{\Phi}^2$ and $\Phi(t) = 1$. Unlike the previous objective $G(\cdot)$, note that our new problem has now a concave objective yet has non-convex (and non closed) constraints. Hence we make a convex relaxation and solve instead

$$\sup_{\Phi \in J_t} \frac{1}{\alpha_\nu - \alpha_\pi}\left(\frac{\alpha_\nu}{2} - \int_0^t \dot{\Phi}_s^2 ds - \frac{\alpha_\nu}{2}\Phi_0^2\right),$$

where $J_t$ is the set of non-negative functions in $W^{1,2}([0, t])$ such that $\alpha_\pi \Phi \leq \dot{\Phi}$ and $\Phi(t) = 1$. After showing that an optimal solution $\Phi$ indeed exists, we explicitly describe it and show that it belongs to $I_t$ a.k.a. it verifies the original constraints. In order to come up with an explicit solution $\Phi$, we make a disjunction of cases on whether the constraint $\alpha_\pi \Phi \leq \dot{\Phi}$ is saturated: if it is always saturated, then $\Phi$ is exponential if not, then it must be linear on a maximal neighborhood around the non saturated point. We then prove that this maximal neighborhood is unique, pathwise connected and can only be of the form $[0, x_0]$ with $x_0 \leq t$ so that $\Phi$ is linear on $[0, x_0]$ and is then exponential on $[x_0, t]$. $\qquad\square$

*Proof of Prop. 6.*  We first notice that minimizing $G_t$ defined in Corollary 5 over the set of admissible tempering schedules $\lambda$ is equivalent to solving

$$\sup_{\lambda \in \Lambda_t} \int_0^t \lambda_s \alpha_s \exp\left(-2\int_s^t \alpha_u du\right) ds,$$

where $\Lambda_t$ is the set of functions from $[0, t]$ to $[0, 1]$ with non-negative weak derivative.

**Proposition 18** *Suppose $\alpha_\pi < \alpha_\nu$ and let $I_t$ be the set of positive functions $\Phi$ with weak second derivative, such that $\Phi_t = 1$, $\alpha_\pi \Phi \leq \dot{\Phi} \leq \alpha_\nu \Phi$ and $\ddot{\Phi}\Phi - \dot{\Phi}^2$ is a non-positive measure. Then*

$$\sup_{\lambda \in \Lambda_t} \int_0^t \lambda_s \alpha_s \exp\left(-2\int \alpha_u du\right) ds = \sup_{\Phi \in I_t} \frac{1}{\alpha_\nu - \alpha_\pi}\left(\frac{\alpha_\nu}{2} - \int_0^t \dot{\Phi}_s^2 ds - \frac{\alpha_\nu}{2}\Phi_0^2\right). \qquad (53)$$

*Proof.* Let $\lambda \in \Lambda_t$ and denote $\Phi_s := \Phi(s) = \exp\left(-\int_s^t \alpha_u du\right)$. The function $\Phi$ is strictly positive and such that $\Phi_t = 1$. Its first and second derivatives read

$$\begin{cases} \dot{\Phi}_s = \alpha_s \Phi_s, \\ \ddot{\Phi}_s = \dot{\alpha}_s \Phi_s + \frac{\dot{\Phi}_s^2}{\Phi_s}. \end{cases}$$

In particular, since $\lambda \in [0,1]$ and $\alpha_s = (1-\lambda_s)\alpha_\nu + \lambda_s \alpha_\pi$, we have $\alpha_s \in [\alpha_\pi, \alpha_\nu]$ and by positiveness of $\Phi$, it holds

$$\alpha_\pi \Phi \le \dot{\Phi} \le \alpha_\nu \Phi.$$

Similarly, since $\dot{\lambda}$ is a non-negative measure, $\dot{\alpha} = (\alpha_\pi - \alpha_\nu)\dot{\lambda}$ is a non-positive measure and so is $\dot{\alpha}\Phi^2$. This implies in particular

$$\ddot{\Phi}\Phi - \dot{\Phi}^2 = \dot{\alpha}\Phi^2 \le 0.$$

Hence we verified that $\Phi \in I_t$. Furthermore, it reads

$$\begin{aligned} \int_0^t \lambda_s \alpha_s \exp\left(-2\int_s^t \alpha_u du\right) ds &= \frac{1}{\alpha_\nu - \alpha_\pi} \int_0^t (\alpha_\nu - \alpha_s)\alpha_s \exp\left(-2\int_s^t \alpha_u du\right) ds \\ &= \frac{1}{\alpha_\nu - \alpha_\pi} \int_0^t \alpha_\nu \dot{\Phi}_s \Phi_s - \dot{\Phi}_s^2 ds \\ &= \frac{1}{\alpha_\nu - \alpha_\pi} \left(\frac{\alpha_\nu}{2}[\Phi_s^2]_0^t - \int_0^t \dot{\Phi}_s^2 ds\right) \\ &= \frac{1}{\alpha_\nu - \alpha_\pi} \left(\frac{\alpha_\nu}{2}(1 - \Phi_0^2) - \int_0^t \dot{\Phi}^2(s) ds\right). \end{aligned}$$

In particular, we have

$$\sup_{\lambda \in \Lambda_t} \int_0^t \lambda_s \alpha_s \exp\left(-2\int \alpha_u du\right) ds \le \sup_{\Phi \in I_t} \frac{1}{\alpha_\nu - \alpha_\pi} \left(\frac{\alpha_\nu}{2} - \int_0^t \dot{\Phi}_s^2 ds - \frac{\alpha_\nu}{2}\Phi_0^2\right).$$

Conversely, let $\Phi \in I_t$, defining $\alpha_s = \partial_s \log \Phi = \dot{\Phi}/\Phi$ and $\lambda_s = \frac{\alpha_\nu - \alpha_s}{\alpha_\pi - \alpha_\nu}$ we have by construction that $\lambda \in \Lambda_t$. Furthermore, the previous computations show that

$$\frac{1}{\alpha_\nu - \alpha_\pi} \left(\frac{\alpha_\nu}{2}(1 - \Phi_0^2) - \int_0^t \dot{\Phi}_s^2 ds\right) = \int_0^t \lambda_s \alpha_s \exp\left(-2\int \alpha_u du\right) ds.$$

In particular, we recover the reverse inequality

$$\sup_{\Phi \in I_t} \frac{1}{\alpha_\nu - \alpha_\pi} \left(\frac{\alpha_\nu}{2} - \int_0^t \dot{\Phi}_s^2 ds - \frac{\alpha_\nu}{2}\Phi_0^2\right) \le \sup_{\lambda \in \Lambda_t} \int_0^t \lambda_s \alpha_s \exp\left(-2\int \alpha_u du\right) ds. \quad \square$$

Hence the problem can be re-written as the minimization of the convex functional $\Phi \mapsto \int_0^t \dot{\Phi}_s^2 ds + \frac{\alpha_\nu}{2}\Phi_0^2$ under the constraint $\Phi \in I_t$. As can be noted, the constraint $\ddot{\Phi}\Phi - \dot{\Phi}^2 \le 0$ is non-convex and the constraints $\Phi > 0$, $\Phi$ has a weak second derivative are non-closed. Hence we are going to relax all of these constraints and solve instead

$$\inf_{\Phi \in J_t} \int_0^t \dot{\Phi}_s^2 ds + \frac{\alpha_\nu}{2}\Phi_0^2, \tag{54}$$

where $J_t \subset W^{1,2}([0,t])$ is the set of positive functions such that $\Phi_t = 1$, $\alpha_\pi \Phi \le \dot{\Phi}$ almost everywhere. Note that we also dropped the constraint $\dot{\Phi} \le \alpha_\nu \Phi$ for convenience of the proof. Nevertheless, we show that at the optimum, the solution of the problem above verifies all the initial constraints.

**Proposition 19** *Assume $\alpha_\pi < \alpha_\nu$. Then Eq. 54 admits a minimizer $\Phi$ which has the following expression:*

 1. *If $\alpha_\pi > \frac{\alpha_\nu}{2}$, then $\Phi_s = e^{\alpha_\pi(s-t)}$.*

2. If $\frac{\alpha_\nu}{t\alpha_\nu+2} < \alpha_\pi \leq \frac{\alpha_\nu}{2}$, then

$$\Phi_s = \begin{cases} \alpha_\pi e^{\alpha_\pi(\frac{1}{\alpha_\pi} - \frac{2}{\alpha_\nu} - t)}s + \frac{2\alpha_\pi}{\alpha_\nu} e^{\alpha_\pi(\frac{1}{\alpha_\pi} - \frac{2}{\alpha_\nu} - t)} & s \leq \frac{1}{\alpha_\pi} - \frac{2}{\alpha_\nu}, \\ e^{\alpha_\pi(s-t)} & s > \frac{1}{\alpha_\pi} - \frac{2}{\alpha_\nu}. \end{cases}$$

3. If $\alpha_\pi \leq \frac{\alpha_\nu}{t\alpha_\nu+2}$, then $\Phi_s = \frac{\alpha_\nu}{2+t\alpha_\nu}s + (1 - \frac{t\alpha_\nu}{2+t\alpha_\nu})$.

*Proof.* We aim to solve the minimization problem defined in Eq. 54.

**Existence of a minimum.** Since the objective has compact sub-level sets and is continuous for the $W^{1,2}([0,t])$ norm, we can extract a minimizing sequence $(\Phi_n)_n$ in $J_t$ that converges in $W^{1,2}$ towards some $\Phi$ in $W^{1,2}([0,t])$. Because $W^{1,2}$ metrizes pointwise convergence, we obtain that the infimum verifies two of the constraints: $\Phi_t = 1$ and $\Phi \geq 0$ almost everywhere.

Then, since $\Phi_n \in J_t$, for $s_0, h > 0$ such that $s_0, s_0 + h \in [0,t]$ we have $0 \leq \alpha_\pi \Phi_n \leq \dot{\Phi}_n$ and then:

$$\alpha_\pi (\Phi_n)_{s_0} \leq \frac{\alpha_\pi}{h} \int_{s_0}^{s_0+h} (\Phi_n)_s \mathrm{d}s \leq \frac{1}{h} \int_{s_0}^{s_0+h} (\dot{\Phi}_n)_s \mathrm{d}s = \frac{(\Phi_n)_{s_0+h} - (\Phi_n)_{s_0}}{h}.$$

Letting $n$ go to infinity and $h$ to zero, we obtain, for almost all $s_0 \in [0,t]$

$$\alpha_\pi \Phi_{s_0} \leq \dot{\Phi}_{s_0},$$

hence $\Phi \in J_t$, showing $\Phi$ is a minimizer of Eq. 54.

**Explicit expression of the minimum.** We now derive an explicit expression for $\Phi$ based on whether or not there exists a point $x_0$ where the constraint

$$\alpha_\pi \Phi \leq \dot{\Phi},$$

is not saturated.

**Case 1: the inequality is saturated everywhere.** In the case where the inequality constraint above is saturated for $x$ almost everywhere, we obtain that the minimizer $\Phi$ is of the form $\Phi_s = K \exp(\alpha_\pi s)$ and using the fact that $\Phi_t = 1$, we recover that $\Phi_s = \exp((s-t)\alpha_\pi)$.

**Case 2: the inequality is not saturated saturated everywhere.** Now let us assume that there exists $x_0$ where the inequality is not saturated by the minimizer

$$\alpha_\pi \Phi(x_0) < \dot{\Phi}(x_0).$$

We can make a disjunction of cases with respect to $x_0$.

**If $x_0 = t$.** We will show that $\Phi$ is linear on $[0,t]$. We first define a neighborhood of $x_0$ of size $\epsilon$ that satisfies a certain inequality. By continuity of $\Phi$ and the definition of the derivative, there exists an $\epsilon > 0$ such that for all $\delta < \epsilon$

$$\alpha_\pi \Phi(t) < \frac{\Phi(t) - \Phi(t-\delta)}{\delta}. \tag{55}$$

Let us now define the neighborhood size $\epsilon$ to be the largest $\epsilon > 0$ such that $t - \epsilon \geq 0$ and such that the inequality above holds for all $\delta < \epsilon$.

Next, we define a candidate $\tilde{\Phi}$ that is linear in this neighborhood, and equal to the minimizer $\Phi$ outside the neighborhood. Let

$$\begin{cases} \tilde{\Phi}(x) = \Phi(x) \text{ if } x \notin [t-\epsilon, t], \\ \tilde{\Phi}(x) = \frac{\Phi(t)-\Phi(t-\epsilon)}{\epsilon}(x - t + \epsilon) + \Phi(t-\epsilon) \text{ if } x \in [t-\epsilon, t]. \end{cases}$$

By construction, $\tilde{\Phi}$ verifies the constraints, i.e. $\tilde{\Phi} \in J_t$. Indeed, taking $\delta \to \epsilon$ in Eq. 55, and using the monotonicity of $\tilde{\Phi}$, we get successively for almost every $x \in [0,t]$:

$$\alpha_\pi \tilde{\Phi}(x) \leq \alpha_\pi \tilde{\Phi}(t) \leq \frac{\Phi(t) - \Phi(t-\epsilon)}{\epsilon} = \dot{\Phi}(x).$$

Hence, $\tilde{\Phi}$ is a suitable candidate for solving Problem Eq. 54.

Finally, we show that the candidate we built $\tilde{\Phi}$ achieves a smaller objective than a minimizer and must therefore be *equal* to a minimizer $\Phi \in J_t$. By optimality,

$$\int_0^t \dot{\Phi}^2(s)ds + \frac{\alpha_\nu}{2}\Phi(0)^2 \leq \int_0^t \dot{\tilde{\Phi}}^2(s)ds + \frac{\alpha_\nu}{2}\tilde{\Phi}(0)^2$$

and since $\Phi$ and $\tilde{\Phi}$ coincide on $[0, t-\epsilon]$, it holds

$$\int_{t-\epsilon}^t \dot{\Phi}^2(s)ds \leq \frac{(\Phi(t) - \Phi(t-\epsilon))^2}{\epsilon}.$$

The same inequality in the other direction is obtained by applying Jensen's inequality

$$\int_{t-\epsilon}^t \dot{\Phi}^2(s)ds \geq \epsilon\left(\frac{1}{\epsilon}\int_{t-\epsilon}^t \dot{\Phi}(s)ds\right)^2 = \frac{(\Phi(t) - \Phi(t-\epsilon))^2}{\epsilon}.$$

We are therefore in the equality case of Jensen with a strongly convex and smooth function, so it must hold for $s$ almost everywhere in the neighborhood $[t - \epsilon, t]$ that

$$\dot{\Phi}(s) = \frac{\Phi(t) - \Phi(t-\epsilon)}{\epsilon},$$

and in particular $\Phi$ is linear in the neighborhood $[t - \epsilon, t]$ and is of the form $\Phi(x) = ax + b$.

Finally, we will show that the neighborhood covers the entire interval $[0, t]$, or in other words, that $\epsilon = t$. Suppose it is not the case and $\epsilon < t$. Then, by continuity of $\Phi$ at the neighborhood border $t - \epsilon$,

$$\alpha_\pi\Phi(t) = \frac{\Phi(t) - \Phi(t-\epsilon)}{\epsilon}.$$

Combined with the fact that $\Phi(t) = 1$, this implies $a = \alpha_\pi$ which is a contradiction with the inequality Eq. 55 that characterizes the neighborhood, and in particular yields $a > \alpha_\pi$. Hence, we must have $\epsilon = t$ and in particular, $\Phi$ is linear on the whole interval $[0, t]$.

**If $x_0 \in ]0, t[$** As before we can define $\epsilon_+$ and $\epsilon_-$ as the largest $\epsilon > 0$ such that $x_0 + \epsilon < t$ (resp. $x_0 - \epsilon > 0$) and $\frac{\Phi(x_0+\delta) - \Phi(x_0)}{\delta} > \alpha_\pi\Phi(x_0+\delta)$ for all $\delta < \epsilon_+$ (resp. $\frac{\Phi(x_0) - \Phi(x_0-\delta)}{\delta} > \alpha_\pi\Phi(x_0)$ for all $\delta < \epsilon_-$). Again, we obtain that $\Phi$ must be linear on $[x_0, x_0 + \epsilon_+]$ of the form $\Phi(x) = a_+ x + b_+$ and linear on $[x_0 - \epsilon_-, x_0]$ of the form $\Phi(x) = a_- x + b_-$. Furthermore, since $\Phi$ is differentiable at $x_0$, its derivative must read

$$\dot{\Phi}(x_0) = \lim_{h\to 0}\frac{\Phi(x_0 + h) - \Phi(x_0)}{h} = \lim_{h\to 0}\frac{\Phi(x_0) - \Phi(x_0 - h)}{h},$$

which implies $a_+ = a_- := a$ and by continuity of $\Phi$ at $x_0$, we also recover $b_+ = b_-$. In particular, $\Phi$ is linear across the whole interval $[x_0 - \epsilon_-, x_0 + \epsilon_+]$. We have now four possibilities: either a) $x_0 - \epsilon_- = 0$ and $x_0 + \epsilon_+ = t$, b) $x_0 - \epsilon_- > 0$ and $x_0 + \epsilon_+ < t$, c) $x_0 - \epsilon_- > 0$ and $x_0 + \epsilon_+ = t$, d) $x_0 - \epsilon_- = 0$ and $x_0 + \epsilon_+ < t$. The case a) simply corresponds to the case where $\Phi$ is linear across the whole interval. The case b) implies by continuity that $a = \alpha_\pi\Phi(x_0) = \alpha_\pi\Phi(x_0 + \epsilon_+)$ which implies that $a = 0$. This cannot hold as we initially assumed that $\alpha_\pi\Phi(x_0) < \dot{\Phi}(x_0) = 0$ which would result in a violation of the positivity constraint. The case c) yields the contradiction $a = \alpha_\pi\Phi(x_0)$ and $a > \alpha_\pi\Phi(x_0)$. Finally, let us deal with case d). We prove that in this case, there exists no other point outside $[0, x_0 + \epsilon_+]$ such that the constraint $\alpha_\pi\Phi \leq \dot{\Phi}$ is not saturated ; in particular this implies that $\Phi$ is linear on $[0, x_0 + \epsilon_+]$ and of the form $K\exp(\alpha_\pi x)$ on $[x_0 + \epsilon_+, t]$. Let us assume that there exists $x_1$ outside of $[0, x_0 + \epsilon_+]$ such that

$$\alpha_\pi\Phi(x_1) < \dot{\Phi}(x_1).$$

Just as before, this implies the existence of $\tilde{\epsilon}_+$ such that $\Phi$ is linear across $[0, x_1 + \tilde{\epsilon}_+]$ and such that

$$\alpha_\pi\Phi(x_1 + \tilde{\epsilon}_+) \leq \frac{\Phi(x_1 + \tilde{\epsilon}_+) - \Phi(x_1)}{\tilde{\epsilon}_+}.$$

By linearity, if we take $\epsilon$ such that $x_0 + \epsilon < x_1 + \tilde{\epsilon}_+$, the right hand side is also given by $\frac{\Phi(x_0+\epsilon)-\Phi(x_0)}{\epsilon}$ and since $\Phi$ is strictly increasing, the left-hand side can be lower-bounded by $\alpha_\pi \Phi(x_0 + \epsilon)$ which implies

$$\alpha_\pi \Phi(x_0 + \epsilon) < \frac{\Phi(x_0 + \epsilon) - \Phi(x_0)}{\epsilon},$$

for all $\epsilon$ such that $x_0 + \epsilon < x_1 + \tilde{\epsilon}_+$. Since $x_1 + \tilde{\epsilon}_+ > x_0 + \epsilon_+$, this contradicts the definiton of $\epsilon_+$ which was chosen as the largest $\epsilon$ such that for all $\epsilon \leq \epsilon_+$,

$$\alpha_\pi \Phi(x_0 + \epsilon) < \frac{\Phi(x_0 + \epsilon) - \Phi(x_0)}{\epsilon}.$$

In particular, the constraint $\alpha_\pi \Phi \leq \dot{\Phi}$ is always saturated outside of $[0, x_0 + \epsilon_+]$.

**If $x_0 = 0$**  As before, we obtain that $\Phi$ is either linear across the whole space $[0, t]$, either there exists $x_0$ such that $\Phi$ is linear across $[0, x_0]$ and of the form $K \exp(\alpha_\pi x)$ on $[x_0, t]$ and such that $\alpha_\pi \Phi(x_0) = \dot{\Phi}(x_0)$.

**Summary of the minimizer candidates**  We can now summarize the candidates we obtained for the minimizer function $\Phi$. It has the three following possible forms:

- Form 1: $\Phi$ is an exponential function of the form $K \exp(\alpha_\pi x)$ on the entire interval $[0, t]$.

- Form 2: $\Phi$ is a linear function on the entire interval $[0, t]$.

- Form 3: $\Phi$ is linear function then an exponential function on the entire interval.

  Specifically, there exists $x_0 \in ]0, t[$ such that $\Phi$ is linear on $[0, x_0]$ and of the form $K \exp(\alpha_\pi x)$ on $[x_0, t]$ and such that $\alpha_\pi \Phi(x_0) = \dot{\Phi}(x_0)$.

We shall determine which candidate $\Phi$ among the three is the true minimizer, by evaluating the objective function Eq. 54 for each and picking the candidate which achieves the lowest value.

**Form 1.**  Since $\Phi(t) = 1$, it implies that the function $\Phi$ is entirely determined and given by $\Phi(x) = \exp(\alpha_\pi(x - t))$. In this case, the value of the objective is given by

$$\int_0^t \dot{\Phi}^2(s)ds + \frac{\alpha_\nu}{2}\Phi(0)^2 = \alpha_\pi^2 \left[\frac{e^{2\alpha_\pi(x-t)}}{2\alpha_\pi}\right]_0^t + \frac{\alpha_\nu}{2}e^{-2\alpha_\pi t}, = \frac{\alpha_\pi}{2} + \frac{e^{-2\alpha_\pi t}}{2}(\alpha_\nu - \alpha_\pi).$$

**Form 2.**  The function $\Phi$ is of the form $\Phi(x) = ax + b$. Since $\Phi(t) = 1$, the coefficient $b$ is given by $1 - at$. Furthermore, the constraint $\alpha_\pi \Phi \leq \dot{\Phi}$ is equivalent to solving $a \geq \alpha_\pi$ and the constraint $\Phi \geq 0$ becomes $a \leq 1/t$. Hence, in the case where $1/t < \alpha_\pi$, there is no linear admissible potential. In the case where, $\alpha_\pi \leq 1/t$, we need to solve the following one dimensional quadratic problem

$$\inf_{1/t \geq a \geq \alpha_\pi} ta^2 + \frac{\alpha_\nu}{2}(1 - at)^2.$$

The objective can be rewritten as $\theta(a) = a^2(t + \frac{t^2\alpha_\nu}{2}) - at\alpha_\nu + \frac{\alpha_\nu}{2}$. The minimum of $\theta$ is given by

$$a_* := \frac{\alpha_\nu}{2 + t\alpha_\nu} \tag{56}$$

Note that we always have $a_* \leq 1/t$, hence if $\alpha_\pi \leq a_*$ the minimum is attained for $a = a_*$ and is given by

$$\theta(a^*) = \frac{\alpha_\nu}{2} - \frac{t\alpha_\nu^2}{4 + 2t\alpha_\nu} = \frac{\alpha_\nu}{2}(1 - \frac{t\alpha_\nu}{2 + t\alpha_\nu}) = \frac{\alpha_\nu}{2 + t\alpha_\nu}.$$

If $\alpha_\pi > a_*$, the minimum is attained for $a = \alpha_\pi$ and is given by

$$\theta(\alpha_\pi) = t\alpha_\pi^2 + \frac{\alpha_\nu}{2}(1 - \alpha_\pi t)^2 = t\alpha_\pi^2 + \frac{\alpha_\nu}{2} - \alpha_\nu\alpha_\pi t + (t\alpha_\pi)^2\frac{\alpha_\nu}{2} = t\alpha_\pi(\alpha_\pi - \alpha_\nu + t\frac{\alpha_\pi\alpha_\nu}{2}) + \frac{\alpha_\nu}{2}.$$

**Form 3.** There exists $0 < x_0 < t$ such that the function $\Phi$ is of the form $ax + b$ over $[0, x_0]$ and then is given by $e^{\alpha_\pi(x-t)}$ over $[x_0, t]$ and which is such that $\dot{\Phi}(x_0) = \alpha_\pi \Phi(x_0)$. The continuity of $\Phi$ implies that $ax_0 + b = e^{\alpha_\pi(x_0-t)}$ and the previous equation gives $a = (ax_0 + b)\alpha_\pi$. Both these equations uniquely determine $a$ and $b$ when $x_0$ is fixed as

$$\begin{cases} a = \alpha_\pi e^{\alpha_\pi(x_0-t)}, \\ b = e^{\alpha_\pi(x_0-t)}(1 - \alpha_\pi x_0), \end{cases}$$

and in particular the positivity constraint is equivalent to $x_0 \leq \frac{1}{\alpha_\pi}$; note that the constraint $\alpha_\pi \Phi \leq \dot{\Phi}$ is indeed respected over $[0, t]$. Hence, we must solve the following one dimensional problem

$$\inf_{0 \leq x_0 \leq \min(t, \frac{1}{\alpha_\pi})} x_0 \alpha_\pi^2 e^{2\alpha_\pi(x_0-t)} + \frac{\alpha_\pi}{2}(1 - e^{2\alpha_\pi(x_0-t)}) + \frac{\alpha_\nu}{2} e^{2\alpha_\pi(x_0-t)}(1 - \alpha_\pi x_0)^2.$$

Let us remark that $x_0 = 0$ being an admissible candidate, the potential $\Phi(x) = e^{\alpha_\pi(x-t)}$ is indeed an admissible candidate and in particular, the value of the problem above is always lower than the one obtained with Form 1. Denoting $f$ the objective function, let us compute the derivative of $f$. For all $x_0$, we have

$$f'(x_0) = \alpha_\pi^2 e^{2(x_0-t)} + 2x_0 \alpha_\pi^3 e^{2(x_0-t)} - \alpha_\pi^2 e^{2(x_0-t)} + \alpha_\pi \alpha_\nu (1 - x_0 \alpha_\pi)^2 e^{2(x_0-t)}$$
$$- \alpha_\nu \alpha_\pi (1 - x_0 \alpha_\pi) e^{2(x_0-t)},$$
$$= e^{2(x_0-t)} \alpha_\pi \Big( 2x_0 \alpha_\pi^2 - \alpha_\nu (1 - x_0 \alpha_\pi)(1 - 1 + x_0 \alpha_\pi) \Big),$$
$$= x_0 e^{2(x_0-t)} \alpha_\pi^2 (2\alpha_\pi - \alpha_\nu (1 - x_0 \alpha_\pi)).$$

The term $(2\alpha_\pi - \alpha_\nu(1 - x_0 \alpha_\pi))$ cancels for $x_0 = \frac{1}{\alpha_\pi} - \frac{2}{\alpha_\nu}$, hence if $\alpha_\pi \geq \frac{\alpha_\nu}{2}$, the derivative $f'$ is always non-negative for all $x_0 \geq 0$ hence the optimal choice of $x_0$ is given by $x_0 = 0$. Let us remark that this case implies $\alpha_\pi > a_*$ where $a_*$ is defined in Eq. 56 and for which the optimal Form 2. is given by $\Phi(x) = \alpha_\pi x + (1 - t\alpha_\pi)$ which an admissibile candidate for Form 3. This proves that whenever $\alpha_\pi > \frac{\alpha_\nu}{2}$, the optimal potential is given by $\Phi(x) = e^{\alpha_\pi(s-t)}$.with $x_0 = t$. If $\alpha_\pi < \frac{\alpha_\nu}{2}$, then $f$ decreases between $0$ and $\frac{1}{\alpha_\pi} - \frac{2}{\alpha_\nu}$ and then increases. In particular, since $\frac{1}{\alpha_\pi} - \frac{2}{\alpha_\nu} < \frac{1}{\alpha_\pi}$, the minimum is attained for $x_0 = \min(t, \frac{1}{\alpha_\pi} - \frac{2}{\alpha_\nu})$. In the case where $t \leq \frac{1}{\alpha_\pi} - \frac{2}{\alpha_\nu}$ which is equivalent to $\alpha_\pi \leq a_*$ with $a_*$ defined in Eq. 56, the obtained potential is linear hence it is necessarily sub-optimal with respect to Form 2. which yields as an optimal potential

$$\Phi(x) = a_* x + (1 - a_* t).$$

Let us now place ourselves in the case $t > \frac{1}{\alpha_\pi} - \frac{2}{\alpha_\nu}$. In the case where $t > \frac{1}{\alpha_\pi} - \frac{2}{\alpha_\nu}$, which is equivalent to $\alpha_\pi > a_*$, the optimal potential yielded by Form 2. (if any) is given by $\Phi(x) = \alpha_\pi x + (1 - \alpha_\pi t)$ which is admissible for Form 3. by taking $x_0 = t$ hence, Form 3. is necessarily sub-optimal.

As a conclusion, we have obtained the following expressions for $\Phi$:

1. If $\alpha_\pi > \frac{\alpha_\nu}{2}$, we have $\Phi(x) = e^{\alpha_\pi(x-t)}$.

2. If $\frac{\alpha_\nu}{t\alpha_\nu+2} < \alpha_\pi \leq \frac{\alpha_\nu}{2}$, then $\Phi$ is given by

$$\begin{cases} \Phi(x) = \alpha_\pi e^{\alpha_\pi(\frac{1}{\alpha_\pi} - \frac{2}{\alpha_\nu} - t)} x + \frac{2\alpha_\pi}{\alpha_\nu} e^{\alpha_\pi(\frac{1}{\alpha_\pi} - \frac{2}{\alpha_\nu} - t)} \text{ if } x \leq \frac{1}{\alpha_\pi} - \frac{2}{\alpha_\nu}, \\ \Phi(x) = e^{\alpha_\pi(x-t)} \text{ if } x > \frac{1}{\alpha_\pi} - \frac{2}{\alpha_\nu}. \end{cases}$$

3. If $\alpha_\pi \leq \frac{\alpha_\nu}{t\alpha_\nu+2}$, then $\Phi(x) = \frac{\alpha_\nu}{2+t\alpha_\nu} x + (1 - \frac{t\alpha_\nu}{2+t\alpha_\nu})$.

Note that in any case, $\Phi$ is in fact continuously differentiable and verifies almost everywhere $\ddot{\Phi}\Phi \leq \dot{\Phi}^2$ as well as $\dot{\Phi} \leq \alpha_\nu \Phi$, hence it is a solution of the problem

$$\inf_{\substack{\Phi \geq 0, \ddot{\Phi}\Phi \leq \dot{\Phi}^2, \\ \Phi(t)=1, \ \alpha_\pi \Phi \leq \dot{\Phi} \leq \alpha_\nu \Phi.}} \int_0^t \dot{\Phi}^2(s)ds + \frac{\alpha_\nu}{2}\Phi(0)^2.$$

$\square$

The optimal $\Phi$ is linked to the optimal tempering scheme $\lambda$ as $\lambda_s = \frac{\alpha_\nu - \alpha_s}{\alpha_\nu - \alpha_\pi}$ with $\alpha_s = \partial_s \log(\Phi)$. Hence we obtain

$$\lambda_s = \min\Big(1, \frac{\alpha_\nu(\alpha_\nu s + 1)}{(\alpha_\nu - \alpha_\pi)(\alpha_\nu s + 2)}\Big).$$

$\square$

### C.3 Proof of Proposition 7

Reversing the integration by parts we have

$$G_t(\lambda) = 1 - 2\int_0^t \lambda_s \alpha_s \exp\Big(-2\int_s^t \alpha_v \mathrm{d}v\Big)\mathrm{d}s = \int_0^t \dot{\lambda}_s \exp\Big(-2\int_s^t \alpha_v \mathrm{d}v\Big)\mathrm{d}s.$$

Hence

$$
\begin{aligned}
G_t(\lambda) &= \int_0^t \dot{\lambda}_s \exp\Big(-2\int_s^t \alpha_v dv\Big)ds \\
&= \frac{1}{t}\int_0^t \exp\Big(-2\int_s^t (\frac{v}{t}(\alpha_\pi - \alpha_\nu) + \alpha_\nu)dv\Big)ds \\
&= \frac{1}{t}\int_0^t \exp\Big(-2\frac{\alpha_\pi - \alpha_\nu}{t}\frac{1}{2}(t^2 - s^2) - 2(t - s)\alpha_\nu\Big)ds \\
&= \frac{1}{t}\int_0^t \exp\Big(-\frac{\alpha_\nu - \alpha_\pi}{t}s^2 + 2\alpha_\nu s - t(\alpha_\nu + \alpha_\pi)\Big)ds \ .
\end{aligned}
$$

Recall that for generic $a, b, c > 0$, we have

$$\int \exp(-ax^2 + bx - c)dx = \frac{\sqrt{\pi}}{2\sqrt{a}}\exp(b^2/4a - c)\,\mathrm{erf}((2ax - b)/(2\sqrt{a})) + \text{constant}.$$

We may apply this formula since $\alpha_\pi < \alpha_\nu$. it yields the equality claim:

$$G_t(\lambda) = \frac{1}{\sqrt{t}}\frac{\sqrt{\pi}}{2\sqrt{\alpha_\nu - \alpha_\pi}}\exp\Big(\frac{t\alpha_\pi^2}{\alpha_\nu - \alpha_\pi}\Big)\Big(\mathrm{erf}\Big(\frac{\sqrt{t}\alpha_\nu}{\sqrt{\alpha_\nu - \alpha_\pi}}\Big) - \mathrm{erf}\Big(\frac{\sqrt{t}\alpha_\pi}{\sqrt{\alpha_\nu - \alpha_\pi}}\Big)\Big) \ .$$

Now, let us consider the behavior of $G_t(\lambda)$ as $t \to \infty$. Here, we Taylor expand to obtain

$$
\begin{aligned}
&\mathrm{erf}\Big(\frac{\sqrt{t}\alpha_\nu}{\sqrt{\alpha_\nu - \alpha_\pi}}\Big) - \mathrm{erf}\Big(\frac{\sqrt{t}\alpha_\pi}{\sqrt{\alpha_\nu - \alpha_\pi}}\Big) \\
&= \sqrt{\frac{\alpha_\nu - \alpha_\pi}{t\pi}}\Big(\frac{1}{\alpha_\pi}\exp\Big(-\frac{t\alpha_\pi^2}{\alpha_\nu - \alpha_\pi}\Big) - \frac{1}{\alpha_\nu}\exp\Big(-\frac{t\alpha_\nu^2}{\alpha_\nu - \alpha_\pi}\Big)\Big) \\
&\quad + O\Big(\frac{1}{t^{3/2}}\Big)\Big(\exp\Big(-\frac{t\alpha_\nu^2}{\alpha_\nu - \alpha_\pi}\Big) - \exp\Big(-\frac{t\alpha_\pi^2}{\alpha_\nu - \alpha_\pi}\Big)\Big) \ .
\end{aligned}
$$

Therefore

$$G_t(\lambda) = \frac{1}{2t\alpha_\pi} - \frac{1}{2t\alpha_\nu}\exp\Big(\frac{-t\alpha_\nu^2 - \alpha_\pi^2}{\alpha_\nu - \alpha_\pi}\Big) + O\Big(\frac{1}{t}\exp\Big(\frac{-t\alpha_\nu^2 - \alpha_\pi^2}{\alpha_\nu - \alpha_\pi}\Big)\Big) \ .$$

This yields the claimed asymptotic result.

These results are numerically validated in Figure 3.

## D Lower bounds

In this section, we give the proofs of Theorems 4, 8 and 9. In section D.1 we prove a general lower bound in total variation for the setup we consider in Theorems 8 and 9. We apply this result to the bimodal example in section D.2. We then study the unimodal examples from Theorems 4 and 9 in section D.3. The bound on the log-Sobolev constant for the uni-modal example is deferred to section D.4. Throughout, we use the notation introduced in section 5.

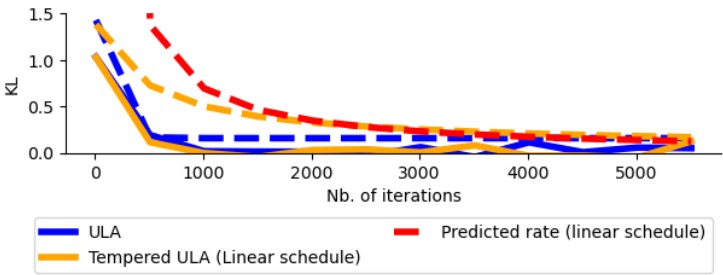

Figure 3: Numerical validation of the rate of convergence predicted by Proposition 7. Dashed lines are our prediction from Proposition 7 and full lines are from simulations of the process using 10 000 particles. The proposal and target are two-dimensional Gaussians, with zero mean and covariance matrices that have a constant diagonal, equal to one for the proposal and 10 for the target. As expected, the predicted rate from Proposition 7 (in dashed red) matches the approximated upper bound from Theorem 3 (in dashed yellow) as well as particle-based simulations (full lines), for large values of time.

### D.1 A GENERAL LOWER BOUND

In this section we state and prove the following general lower bound for the geometric path; in the following sections it is applied to obtain Theorems 8 and 9.

**Theorem 20** (Total variation lower bounds for the geometric path) *Suppose $\nu, \pi \in \mathcal{P}_{ac}(\mathbb{R})$ have log-Lipschitz densities and there is an interval $I = [a, b] \subset \mathbb{R}$ and $\lambda_\star \in [0, 1]$, $\delta, B > 0$ such that:*

1. *Bounded scores on $I$: both $|V'_\nu(x)|, |V'_\pi(x)| \leq B$ for $x \in I$.*

2. *Small mass on $I$: for all $\lambda \in [0, \lambda_\star]$, we have $\mu_\lambda(I) \leq \delta$.*

*Then for each $k \in [K]$ such that $\lambda_k \leq \lambda_\star$,*

$$\mathrm{TV}(p_{T_k}^k, \pi) \geq \pi([b, \infty)) - \pi(I) - \nu([b, \infty)) - \delta - \frac{B}{b - a} \cdot \sqrt{\delta} \cdot \left( \sum_{i=1}^k T_i (\chi^2(p_0^i, \mu_i) + 1)^{1/2} \right).$$

This result will be typically be applied in conjuction with the following lemma, which allows us to control the $\chi^2$ terms arising above when we have pointwise comparisons of one distribution to another.

**Lemma 21** *Suppose that there exists a $C > 0$ such that in the support of $\pi$, $\nu(x) \leq C\pi(x)$. Then*

$$\chi^2(p_0^k, \mu_k) \leq C^{\lambda_k} - 1.$$

*Proof of Lemma 21.* Take $k \in [K]$. Write

$$\chi^2(p_0^k, \mu_k) = \int \frac{(p_0^k)^2}{\mu_k} - 1 = \int \frac{(p_0^k)^2}{\mu_{k-1}} \cdot \frac{\mu_{k-1}}{\mu_k} - 1.$$

Now, letting $c_i := c_{\lambda_i}$ for each $i \in \{0, \ldots, k\}$, we have

$$\frac{\mu_{k-1}}{\mu_k} = \frac{c_{k-1}}{c_k} \cdot \nu^{\lambda_k - \lambda_{k-1}} \pi^{\lambda_{k-1} - \lambda_k} \leq C^{\lambda_k - \lambda_{k-1}} \cdot \frac{c_{k-1}}{c_k}.$$

Hence

$$\chi^2(p_0^k, \mu_k) \leq C^{\lambda_k - \lambda_{k-1}} \cdot \frac{c_{k-1}}{c_k} \int \frac{(p_0^k)^2}{\mu_{k-1}} - 1$$

$$= C^{\lambda_k - \lambda_{k-1}} \cdot \frac{c_{k-1}}{c_k} \int \frac{(p_{T_{k-1}}^{k-1})^2}{\mu_{k-1}} - 1.$$

But since $\chi^2(\cdot, \mu_{k-1})$ is non-increasing under Langevin dynamics towards $\mu_{k-1}$, we find that

$$\chi^2(p_0^k, \mu_k) \leq C^{\lambda_k - \lambda_{k-1}} \cdot \frac{c_{k-1}}{c_k} \int \frac{(p_0^{k-1})^2}{\mu_{k-1}} - 1$$

Proceeding recursively, we obtain

$$\chi^2(p_0^k, \mu_k) \leq C^{\lambda_k} \cdot \frac{c_0}{c_k} \int \frac{(p_0^0)^2}{\mu_0} - 1$$
$$= C^{\lambda_k} \cdot \frac{c_0}{c_k} - 1 = C^{\lambda_k} \cdot \frac{1}{c_k} - 1 \leq C^{\lambda_k} - 1,$$

where at the last step we use the fact that for all $\lambda \in [0, 1]$, Jensen's inequality implies that

$$\frac{1}{c_\lambda} = \int \nu^{1-\lambda}(x)\pi^\lambda(x)\mathrm{d}x \leq 1.$$

$\square$

*Proof.* For convenience, let us put

$$V_\lambda(x) := -\log \mu_\lambda(x), \qquad \lambda \in [0, 1].$$

Define

$$\psi(x) := \begin{cases} 0 & x < a \\ \frac{1}{b-a}(x-a) & x \in I \\ 1 & x > b. \end{cases}$$

We write

$$\pi(\psi) - p_{T_k}^k(\psi) = \pi(\psi) - \nu(\psi) + \sum_{i=1}^{i} p_{T_{i-1}}^{i-1}(\psi) - p_{T_i}^i(\psi).$$

Observe

$$\pi(\psi) \geq \pi([b, \infty)).$$

On the other hand,

$$\nu(\psi) \leq \nu([a, \infty)) = \nu(I) + \nu([b, \infty)) \leq \delta + \nu([b, \infty)).$$

We thus find

$$\pi(\psi) - p_{T_k}^k(\psi) \geq \pi([b, \infty)) - \nu([b, \infty)) - \delta + \sum_{i=1}^{k} p_{T_{i-1}}^{i-1}(\psi) - p_{T_i}^i(\psi).$$

To reduce to a lower bound in total variation, we note that

$$\pi(\psi) - p_{T_k}^k(\psi) \leq \pi([a, \infty)) - p_{T_k}^k([b, \infty)) \leq \mathrm{TV}(p_{T_k}^k, \pi) + \pi(I)$$

We finally obtain the lower bound

$$\mathrm{TV}(p_{T_k}^k, \pi) \geq \pi([b, \infty)) - \pi(I) - \nu([b, \infty)) - \delta + \sum_{i=1}^{k} p_{T_{i-1}}^{i-1}(\psi) - p_{T_i}^i(\psi). \qquad (57)$$

To conclude from Eq. 57, we study the above summands. Notice that by definition, $p_{T_{i-1}}^{i-1}(\psi) = p_0^i(\psi)$, so that these terms are the difference of the expectation of $\psi$ at the start and end of Langevin dynamics run towards $\mu_i$. Hence,

$$p_{T_{k-1}}^{k-1}(\psi) - p_{T_k}^k(\psi) = -\int_0^{T_k} \partial_t \langle \psi, p_t^k \rangle_{L^2(\mu_k)} \mathrm{d}t.$$

Now, $p_t^k$ evolves by the weighted Fokker-Planck operator

$$\partial_t p_t^k = \mathcal{L}_{\mu_k} p_t^k := \Delta p_t^k - \langle \nabla V_{\lambda_k}, \nabla p_t^k \rangle,$$

so by self-adjointness of $\mathcal{L}_{\mu_k}$, we obtain

$$\int_0^{T_k} \partial_t \langle \psi, p_t^k \rangle_{L^2(\mu_k)} \mathrm{d}t = \int_0^{T_k} \langle \psi, \mathcal{L}_{\mu_k} p_t^k \rangle_{L^2(\mu_k)} \mathrm{d}t$$
$$= \int_0^{T_k} \langle \mathcal{L}_{\mu_k} \psi, p_t^k \rangle_{L^2(\mu_k)} \mathrm{d}t.$$

Notice that since $\psi$ is linear where it is non-constant, we have

$$\mathcal{L}_{\mu_k} \psi(x) = -\langle \nabla V_{\lambda_k}(x), \nabla \psi(x) \rangle = \begin{cases} -\frac{1}{b-a} \partial_x V_{\lambda_k}(x) & x \in I \\ 0 & x \notin I. \end{cases}$$

Hence

$$\langle \mathcal{L}_{\mu_k} \psi, p_t^k \rangle_{L^2(\mu_k)} = \frac{-1}{b-a} \int_a^b \partial_x V_{\lambda_k}(x) p_t^k(x) \mathrm{d}\mu_k(x)$$
$$\leq \frac{B}{b-a} \int_a^b p_t^k(x) \mathrm{d}\mu_k(x).$$

Now, we use Cauchy-Schwarz to observe that

$$\int_a^b p_t^k(x) \mathrm{d}\mu_k(x) \leq \mu_k(I)^{1/2} \Big( \int_a^b p_t^k(x)^2 \mathrm{d}\mu_k(x) \Big)^{1/2}$$
$$\leq \mu_k(I)^{1/2} (\chi^2(p_t^k, \mu_k) + 1)^{1/2}$$
$$\leq \sqrt{\delta} (\chi^2(p_t^k, \mu_k) + 1)^{1/2}$$
$$\leq \sqrt{\delta} (\chi^2(p_0^k, \mu_k) + 1)^{1/2},$$

where the last step follows because $\chi^2(\cdot, \mu_k)$ is non-increasing along Langevin dynamics towards $\mu_k$. Hence

$$p_{T_{k-1}}^{k-1}(\psi) - p_{T_k}^k(\psi) = -\int_0^{T_k} \langle \mathcal{L}_{\mu_k} \psi, p_t^k \rangle_{L^2(\mu_k)} \geq -\frac{B T_k}{b-a} \sqrt{\delta} (\chi^2(p_0^k, \mu_k) + 1)^{1/2}.$$

Plugging this inequality into Eq. 57 yields the result. $\qquad \square$

## D.2 LOWER BOUNDS FOR BIMODAL TARGET

In this section, we apply the general lower bound from Theorem 20 to prove Theorem 8, and accordingly throughout take $\nu$ and $\pi$ to be as defined there. We begin by collecting some useful facts about the geometric path between $\nu$ and $\pi$; the proof is included after that of Theorem 8.

**Proposition 22** (Useful facts about the bi-modal example.) *Let $\nu, \pi$ be as in Theorem 8 for $m \geq 11$. Then*

1. *for all $\lambda \in [0, 1]$, the normalizing constant $c_\lambda \in [1, 2]$.*

2. *for each $\lambda \in [0, 1]$,*
$$\mu_\lambda([m/4, 3m/4]) \leq 6e^{-m^2/32}.$$

3. *for all $k \in [K]$ and $t \in [0, T_k]$,*
$$\chi^2(p_t^k, \mu_k) \leq 1.$$

*Proof of Theorem 8.* We will apply Theorem 20 with $I = [m/4, 3m/4]$. An easy calculation shows that we may take $B = 2m$ for the first assumption, and Proposition 22 shows that $\lambda_0 = 1$ and $\delta = 6e^{-m^2/32}$ suffice for the second assumption. We thus obtain

$$\mathrm{TV}(p_{T_k}^k, \pi) \geq \pi([3m/4, \infty)) - \nu([3m/4, \infty)) - 12e^{-m^2/32} - 4\sqrt{6} \Big( \sum_{i=1}^k T_i(\chi^2(p_0^i, \mu_i) + 1)^{1/2} \Big) e^{-m^2/64}.$$

Applying Proposition 22 to control the $\chi^2$ terms yields

$$\text{TV}(p_{T_k}^k, \pi) \geq \pi([3m/4, \infty)) - \nu([3m/4, \infty)) - 12e^{-m^2/32} - 16\Big(\sum_{i=1}^k T_i\Big)e^{-m^2/32}$$

Note that by Mill's ratio, we can bound using $m \geq 4$ that

$$\nu([3m/4, \infty)) \leq e^{-9m^2/32} \leq e^{-m^2/32}.$$

On the other hand,

$$\begin{aligned}
\pi([3m/4, \infty)) &= \pi([m/2, \infty)) - \pi([m/2, 3m/4]) \\
&\geq \pi([m/2, \infty)) - \pi([m/4, 3m/4]) \\
&= \frac{1}{2} - \pi([m/4, 3m/4]) \\
&\geq \frac{1}{2} - 6e^{-m^2/32},
\end{aligned}$$

where at the last step we applied Proposition 22 one more time. Plugging in terms yields

$$\text{TV}(p_{T_k}^k, \pi) \geq \frac{1}{2} - 19e^{-m^2/32} - 16\Big(\sum_{i=1}^k T_i\Big)e^{-m^2/32}.$$

Now, we use our assumption that $m \geq 11$ to observe that

$$\frac{1}{2} - 19e^{-m^2/32} \geq \frac{1}{20},$$

and finally conclude. $\qquad\square$

*Proof of Proposition 22. Proof of Fact 1.* Note that Cauchy-Schwarz immediately implies that for any $\lambda \in [0, 1]$,

$$\frac{1}{c_\lambda} = \int \nu^{1-\lambda}\pi^\lambda \leq 1.$$

On the other hand, using the fact that $2\pi \geq \nu$, we have that

$$\frac{1}{c_\lambda} = \int \nu^{1-\lambda}\pi^\lambda = \int \Big(\frac{\pi}{\nu}\Big)^\lambda \nu \geq 2^{-\lambda} \geq \frac{1}{2}.$$

The claim about the normalizing constant follows.

*Proof of Fact 2.* We use the previous part to observe

$$\begin{aligned}
\mu_\lambda([m/4, 3m/4]) &\leq 2\int_{m/4}^{3m/4} \nu^{1-\lambda}(x)\pi^\lambda(x)\mathrm{d}x \\
&\leq 2\nu([m/4, 3m/4])^{1-\lambda}\pi([m/4, 3m/4])^\lambda \\
&= 2\nu([m/4, 3m/4]),
\end{aligned}$$

where the last equality follows by symmetry. We then observe that by Mills' ratio,

$$\nu([m/4, 3m/4]) \leq \nu([m/4, \infty)) \leq 3e^{-m^2/32}.$$

The result follows.

*Proof of Fact 3.* The result follows immediately from Lemma 21 once we observe that $\nu \leq 2\pi$. $\quad\square$

## D.3 LOWER BOUNDS FOR UNIMODAL TARGET

In this section, we prove Theorem 4 as well as Theorem 9. Since these results concern the same distributions with different mixture weights, in this section we generalize slightly to consider, for some $a \in [\sqrt{2\log 2}/m, 1]$, the target

$$\pi = (1 - e^{-a^2 m^2/2})\mathcal{N}(m, 1) + e^{-a^2 m^2/2}u_m, \tag{58}$$

By taking $a = 1/\sqrt{2}$, we recover the setting of Theorem 4, and by taking $a = \sqrt{2\log 2}/m$ we recover the setting of Theorem 9. Note that when dealing with various numerical constants, we will frequently use the assumption $m \geq 10$ without comment. For both results, the following facts will be useful.

**Proposition 23** (Useful facts about the unimodal target) *Let $\nu := \mathcal{N}(0, 1)$ and $\pi$ be as in Eq. 58 for $m \geq 10$. Then*

1. *For all $\lambda \in [0, 1]$,*

$$\mu_\lambda([m(1-a)/2, m(1-a)]) \leq 5am^2 c_\lambda e^{-\lambda a^2 m^2/2 - (1-\lambda)(1-a)^2 m^2/8}.$$

2. *If $1 \geq a + 2/m$, then for all $\lambda \in [0, 1]$,*

$$\mu_\lambda((-\infty, m(1-a)/2]) \geq \frac{c_\lambda e^{-\lambda a^2 m^2/2}}{10m}.$$

3. *For all $\lambda$ such that $1 \leq a + 2\lambda - 2/m$, we have*

$$\mu_\lambda([m(1-a)/2, \infty)) \geq \frac{c_\lambda}{2} e^{-\frac{1}{2}\lambda(1-\lambda)m^2}.$$

4. *For all $\lambda \in [0, 1]$*

$$\frac{1}{4(e^{-\lambda a^2 m^2/2} + e^{-\frac{1}{2}\lambda(1-\lambda)m^2})} \leq c_\lambda \leq 10m e^{\lambda a^2 m^2/2}.$$

We also use the following.

**Lemma 24** (Log-Sobolev constants of the unimodal target.) *Let $\pi$ be as in Eq. 58 for $m \geq 10$. Then*

$$C_{\mathrm{LS}}(\pi) \leq \frac{81a^2 m^5}{1 - 2e^{-a^2 m^2/2}},$$

*with the convention that the right-hand side is $324m^3$ when $a = \sqrt{2\log 2}/m$.*

The proof of Lemma 24 requires a digression into log-Sobolev inequalities for mixtures and restrictions, so is deferred to the next section, Appendix D.4.

We first prove Theorems 4 and 9, then return to proving Proposition 23.

*Proof of Theorem 4.* The fact that $C_{LS}(\nu) = 1$ is the Gaussian log-Sobolev inequality, and the bound $C_{LS}(\pi) \leq 6m^5$ follows immediately from Lemma 24 upon plugging in $a = 1/\sqrt{2}$.

For the lower bound on $C_P(\mu_\lambda)$, take $\lambda \in [\frac{1}{2}, 1]$ and let $I := [m(1-a)/2, m(1-a)]$; recall that $a \in [0, 1)$ is a free parameter controlling the mixture weights in Eq. 58. Put

$$\psi(x) := \begin{cases} 1 & x < m(1-a)/2, \\ 1 - \frac{2}{(1-a)m}(x - m(1-a)/2) & x \in I, \\ 0 & x > m(1-a). \end{cases}$$

Then

$$\|\nabla \psi\|^2_{L^2(\mu_\lambda)} = \frac{4}{m^2(1-a)^2}\mu_\lambda(I).$$

On the other hand,

$$\begin{aligned}
\mathrm{Var}_{\mu_\lambda}(\psi) &= \mu_\lambda(\psi^2) - \mu_\lambda(\psi)^2 \\
&\geq \mu_\lambda((-\infty, m(1-a)/2]) - (\mu_\lambda((-\infty, m(1-a)/2]) + \mu_\lambda(I))^2 \\
&= \mu_\lambda((-\infty, m(1-a)/2])(1 - \mu_\lambda((-\infty, m(1-a)/2]))) \\
&\quad - 2\mu_\lambda(I)\mu_\lambda((-\infty, m(1-a)/2]) - \mu_\lambda(I)^2 \\
&\geq \mu_\lambda((-\infty, m(1-a)/2])(1 - \mu_\lambda((-\infty, m(1-a)/2]))) - 3\mu_\lambda(I) \\
&= \mu_\lambda((-\infty, m(1-a)/2])\mu_\lambda([m(1-a)/2, \infty)) - 3\mu_\lambda(I)
\end{aligned}$$

We obtain

$$C_P(\mu_\lambda) \geq \frac{\text{Var}_{\mu_\lambda}(\psi)}{\|\nabla\psi\|_{L^2(\mu_\lambda)}^2}$$

$$= \frac{m^2(1-a)^2}{4}\mu_\lambda(I)^{-1}\mu_\lambda((-\infty, m(1-a)/2])\mu_\lambda([m(1-a)/2, \infty)) - \frac{3m^2(1-a)^2}{4}. \quad (59)$$

We'll now apply Proposition 23 with $a = 1/\sqrt{2}$. With this choice of $a$ and for $\lambda \in [\frac{1}{2}, 1]$, the sufficient conditions are all verified, so that

$$\mu_\lambda(I)^{-1}\mu_\lambda((-\infty, m(1-a)/2])\mu_\lambda([m(1-a)/2, \infty)) \geq \frac{1}{100am^3}c_\lambda e^{(1-\lambda)m^2(1-a)^2/8 - \frac{1}{2}\lambda(1-\lambda)m^2}.$$

Applying Proposition 23 once more, this time to control $c_\lambda$, we obtain

$$\mu_\lambda(I)^{-1}\mu_\lambda((-\infty, m(1-a)/2])\mu_\lambda([m(1-a)/2, \infty)) \geq \frac{e^{(1-\lambda)m^2(1-a)^2/8 - \frac{1}{2}\lambda(1-\lambda)m^2}}{400am^3(e^{-\lambda a^2 m^2/2} + e^{-\frac{1}{2}\lambda(1-\lambda)m^2})}.$$

To complete the proof of Theorem 4, we finally plug in $a = 1/\sqrt{2}$, and observe that for $\lambda \in [\frac{1}{2}, 1]$ it holds that $e^{-\lambda a^2 m^2/2} \leq e^{-\frac{1}{2}\lambda(1-\lambda)m^2}$, so that

$$\mu_\lambda(I)^{-1}\mu_\lambda((-\infty, m(1-a)/2])\mu_\lambda([m(1-a)/2, \infty)) \geq \frac{1}{800m^3}e^{(1-\lambda)m^2/100}.$$

Plugging into Eq. 59 we find

$$C_P(\mu_\lambda) \geq \frac{m^2(1-1/\sqrt{2})^2}{4}\left(\frac{1}{800m^3}e^{(1-\lambda)m^2/100} - 3\right).$$

Observe that $\frac{1}{12} \leq (1 - 1/\sqrt{2})^2 \leq 1$, so that

$$C_P(\mu_\lambda) \geq \frac{1}{48 \cdot 800m}e^{(1-\lambda)m^2/100} - \frac{3m^2}{4} \geq \frac{1}{4 \cdot 10^4 m}e^{(1-\lambda)m^2/100} - m^2.$$

$$\square$$

*Proof of Theorem 9.* The equality $C_{LS}(\nu) = 1$ is the Gaussian log-Sobolev inequality. For the log-Sobolev constant of $\pi$, the result is immediate from Lemma 24 upon plugging in $a = \sqrt{2\log 2}/m$.

For the lower bound in total variation, we will apply Theorem 20 using the interval $I = [m(1 - a)/2, m(1 - a)]$; recall that $a \in [0, 1)$ is a free parameter controlling the mixture weights in Eq. 58 that we will eventually set to $a = \sqrt{2\log 2}/m$.

In this case, it clear that we may bound the scores with $B = m$. The main issue is then to get control on $\mu_\lambda(I)$, and for this we apply Proposition 23 to obtain

$$\mu_\lambda(I) \leq 50am^3 e^{-(1-\lambda)(1-a)^2 m^2/8}.$$

So for $\lambda_\star := \lambda_k$, we may take

$$\delta := 50am^3 e^{-(1-\lambda_\star)(1-a)^2 m^2/8}.$$

Theorem 20 then implies

$$\text{TV}(p_{T_k}^k, \pi) \geq \pi([m(1-a), \infty)) - \pi([m(1-a)/2, m(1-a)]) - \nu([m(1-a), \infty)) - \delta$$

$$- \frac{2}{1-a} \cdot \sqrt{\delta} \cdot \left(\sum_{i=1}^{k} T_i(\chi^2(p_0^i, \mu_i) + 1)^{1/2}\right).$$

Next, we observe that $4mu_m(x) \geq \nu(x)$ for all $x \in \mathbb{R}$, so that

$$\pi(x) \geq \frac{1}{2}u_m(x) \geq \frac{1}{8m}\nu(x).$$

Hence we may Lemma 21 with $C = 8m$ to obtain

$$\text{TV}(p_{T_k}^k, \pi) \geq \pi([m(1-a), \infty)) - \pi([m(1-a)/2, m(1-a)]) - \nu([m(1-a), \infty))$$
$$- \delta - \frac{8\sqrt{m}}{1-a} \cdot \sqrt{\delta} \cdot \left(\sum_{i=1}^{k} T_i\right).$$

Let us specialize to $a = \sqrt{2\log 2}/m$. In this case, Mills' ratio implies

$$\nu([m(1-a), \infty)) \leq e^{-(1-a)^2 m^2/2} \leq e^{-2m^2/5}.$$

For the other two terms, let us adopt the notation $g_m := \frac{1}{\sqrt{2\pi}} e^{-\frac{1}{2}(x-m)^2}$, and then write

$$\pi([m(1-a), \infty)) - \pi([m(1-a)/2, m(1-a)])$$
$$\geq \int_{m(1-a)}^{2m} \left(\frac{1}{2} g_m(x) + \frac{1}{2} u_m(x)\right) \mathrm{d}x - \int_{m(1-a)/2}^{m(1-a)} \left(\frac{1}{2} g_m(x) + \frac{1}{2} u_m(x)\right) \mathrm{d}x$$
$$= \frac{1}{2} \int_{m(1-a)}^{2m} g_m(x) \mathrm{d}x - \frac{1}{2} \int_{m(1-a)/2}^{m(1-a)} g_m(x) \mathrm{d}x + \frac{1}{2} \int_{m(1-a/2)}^{2m} u_m(x) \mathrm{d}x$$
$$\geq \frac{1}{2} \int_{m(1-a)}^{2m} g_m(x) \mathrm{d}x - \frac{1}{2} \int_{m(1-a)/2}^{m(1-a)} g_m(x) \mathrm{d}x$$
$$= \int_{m(1-a)}^{m(1+a)} g_m(x) \mathrm{d}x \geq \int_{m-1}^{m+1} g_m(x) \geq \frac{1}{4}.$$

We finally obtain

$$\text{TV}(p_{T_k}^k, \pi) \geq \frac{1}{4} - e^{-2m^2/5} - \delta - 10m \cdot \sqrt{\delta} \cdot \left(\sum_{i=1}^{k} T_i\right).$$

For this choice of $a$, note that

$$\delta \leq 6m^3 e^{-(1-\lambda_k)m^2/10}.$$

By the restriction on $m$ we know $1/4 - e^{-2m^2/5} \geq 1/5$, yielding the result. $\qquad \square$

*Proof of Prop. 23.* For ease of notation, let us write $g_m(x) := \frac{1}{\sqrt{2\pi}} e^{-\frac{1}{2}(x-m)^2}$. Also, put $u_m(x) = c_{u_m} e^{-\frac{1}{2} d(x, I_m)^2}$ for $c_{u_m}$ a normalizing constant and observe that

$$c_{u_m}^{-1} = \int_{-\infty}^{\infty} e^{-\frac{1}{2} d(x, I_m)^2} \mathrm{d}x = 3m + \sqrt{2\pi} \implies \frac{1}{4m} \leq c_{u_m} \leq \frac{1}{3m}.$$

Therefore,

$$e^{-a^2 m^2/2} u_m(x) \leq \frac{e^{-a^2 m^2/2}}{3m} \leq \frac{1}{m} g_m(x), \qquad \forall x \in [m(1-a), m(1+a)], \qquad (60)$$

and

$$g_m(x) \leq e^{-a^2 m^2/2} \leq 4m e^{-a^2 m^2/2} u_m(x) \qquad \forall x \in [-m, m(1-a)] \cup [m(1+a), 2m]. \quad (61)$$

*Proof of Fact 1.* We use Eq. 61 to observe that

$$\mu_\lambda([m(1-a)/2, m(1-a)]) = c_\lambda \int_{m(1-a)/2}^{m(1-a)} \nu^{1-\lambda} \pi^\lambda$$
$$\leq c_\lambda (5m)^\lambda e^{-\lambda a^2 m^2/2} \int_{m(1-a)/2}^{m(1-a)} \nu^{1-\lambda}$$
$$\leq 5am^2 c_\lambda e^{-\lambda a^2 m^2/2} \cdot e^{-(1-\lambda)m^2(1-a)^2/8}.$$

*Proof of Fact 2.* We compute

$$
\begin{aligned}
\mu_\lambda((-\infty, m(1-a)/2]) &= \int_{-\infty}^{m(1-a)/2} c_\lambda \nu^{1-\lambda} \pi^\lambda \mathrm{d}x \\
&\geq c_\lambda \int_{-m(1-a)/2}^{m(1-a)/2} \nu^{1-\lambda} \pi^\lambda \mathrm{d}x \\
&\geq c_\lambda \left(\frac{e^{-a^2 m^2/2}}{4m}\right)^\lambda \left(\frac{1}{\sqrt{2\pi}}\right)^{1-\lambda} \int_{-m(1-a)/2}^{m(1-a)/2} e^{-\frac{1}{2}x^2(1-\lambda)} \mathrm{d}x \\
&\geq c_\lambda \frac{e^{-\lambda a^2 m^2/2}}{4\sqrt{2\pi}m} \int_{-m(1-a)/2}^{m(1-a)/2} e^{-\frac{1}{2}x^2(1-\lambda)} \mathrm{d}x \\
&\geq c_\lambda \frac{e^{-\lambda a^2 m^2/2}}{4\sqrt{2\pi}m} \int_{-m(1-a)/2}^{m(1-a)/2} e^{-\frac{1}{2}x^2} \mathrm{d}x \\
&\geq c_\lambda \frac{e^{-\lambda a^2 m^2/2}}{4\sqrt{2\pi}m} \int_{-1}^{1} e^{-\frac{1}{2}x^2} \mathrm{d}x \\
&\geq c_\lambda \frac{2e^{-1/2}e^{-\lambda a^2 m^2/2}}{4\sqrt{2\pi}m} \\
&\geq \frac{c_\lambda e^{-\lambda a^2 m^2/2}}{10m}.
\end{aligned}
$$

*Proof of Fact 3.* We calculate,

$$
\begin{aligned}
\mu_\lambda([m(1-a)/2, \infty)) &= \int_{m(1-a)/2}^{\infty} c_\lambda \nu^{1-\lambda} \pi^\lambda \mathrm{d}x \\
&\geq \frac{c_\lambda}{2} \int_{m(1-a)/2}^{2m} \frac{1}{\sqrt{2\pi}} e^{-\frac{1}{2}(1-\lambda)x^2 - \frac{1}{2}\lambda(x-m)^2} \mathrm{d}x \\
&= \frac{c_\lambda}{2} e^{-\frac{1}{2}\lambda(1-\lambda)m^2} \int_{m(1-a)/2}^{2m} \frac{1}{\sqrt{2\pi}} e^{-\frac{1}{2}(x-\lambda m)^2} \mathrm{d}x \\
&\geq \frac{c_\lambda}{2} e^{-\frac{1}{2}\lambda(1-\lambda)m^2},
\end{aligned}
$$

where we used the fact that under the hypothesis, we have $m(1-a)/2 \leq \lambda m - 1$.

*Proof of Fact 4.* We compute

$$
\begin{aligned}
\frac{1}{c_\lambda} = \int_{-\infty}^{\infty} \nu^{1-\lambda} \pi^\lambda &\geq \frac{1}{(2\pi)^{(1-\lambda)/2}} \cdot \left(\frac{e^{-a^2 m^2/2}}{4m}\right)^\lambda \cdot \int_{-m/2}^{m/2} e^{-\frac{1}{2}x^2(1-\lambda)} \mathrm{d}x \\
&\geq \frac{e^{-\lambda a^2 m^2/2}}{4\sqrt{2\pi}m} \int_{-m/2}^{m/2} e^{-\frac{1}{2}x^2(1-\lambda)} \mathrm{d}x \\
&\geq \frac{e^{-\lambda a^2 m^2/2}}{4\sqrt{2\pi}m} \int_{-m/2}^{m/2} e^{-\frac{1}{2}x^2} \mathrm{d}x \\
&\geq \frac{2e^{-1/2}e^{-\lambda a^2 m^2/2}}{4\sqrt{2\pi}m} \\
&\geq \frac{e^{-\lambda a^2 m^2/2}}{10m}.
\end{aligned}
$$

On the other hand, we use Eq. 60 and Eq. 61 to bound

$$
\frac{1}{c_\lambda} = \int_{-\infty}^{\infty} \nu^{1-\lambda} \pi^\lambda
$$

$$
= \int_{-m}^{m(1-a)} \nu^{1-\lambda} \pi^\lambda + \int_{m(1-a)}^{m(1+a)} \nu^{1-\lambda} \pi^\lambda + \int_{m(1+a)}^{2m} \nu^{1-\lambda} \pi^\lambda
$$

$$
+ \int_{-\infty}^{-m} \nu^{1-\lambda} \pi^\lambda + \int_{2m}^{\infty} \nu^{1-\lambda} \pi^\lambda
$$

$$
\leq 2 \int_{-m}^{m(1-a)} \nu^{1-\lambda} \pi^\lambda + \int_{m(1-a)}^{m(1+a)} \nu^{1-\lambda} \pi^\lambda + \int_{-\infty}^{-m} \nu^{1-\lambda} \pi^\lambda + \int_{2m}^{\infty} \nu^{1-\lambda} \pi^\lambda
$$

$$
\leq 2 \Big(\frac{5 e^{-am^2/2}}{3}\Big)^\lambda \cdot \frac{1}{(2\pi)^{(1-\lambda)/2}} \int_{-m}^{m(1-a)} e^{-\frac{1}{2}(1-\lambda)x^2} \mathrm{d}x
$$

$$
+ \frac{2^\lambda}{\sqrt{2\pi}} \int_{m(1-a)}^{m(1+a)} e^{-\frac{1}{2}(1-\lambda)x^2 - \frac{1}{2}\lambda(x-m)^2} \mathrm{d}x + \int_{-\infty}^{-m} \nu^{1-\lambda} + \int_{2m}^{\infty} \nu^{1-\lambda}
$$

$$
\leq 4 e^{-\lambda a^2 m^2/2} + \frac{2}{\sqrt{2\pi}} e^{-\frac{1}{2}\lambda(1-\lambda)m^2} \int_{-\infty}^{\infty} e^{-\frac{1}{2}(x-\lambda m)^2} \mathrm{d}x + \frac{2}{\sqrt{2\pi}} \int_{-\infty}^{-m} e^{-\frac{1}{2}x^2} \mathrm{d}x
$$

$$
\leq 4 e^{-\lambda a^2 m^2/2} + 2 e^{-\frac{1}{2}\lambda(1-\lambda)m^2} + e^{-\frac{1}{2}m^2}
$$

$$
\leq 4 \big(e^{-\lambda a^2 m^2/2} + e^{-\frac{1}{2}\lambda(1-\lambda)m^2}\big). \qquad \square
$$

### D.4 Upper bounds on the log-Sobolev constant of the unimodal target

In this section we prove Lemma 24 on the log-Sobolev constant of the unimodal target defined in Eq. 17. We use the following result which controls the log-Sobolev constants of mixtures, and appears as (Schlichting, 2019, Corollary 2).

**Theorem 25** (Upper bounds on log-Sobolev constant of mixtures (Schlichting, 2019)) *Suppose $Q_0, Q_1 \in \mathcal{P}(\mathbb{R}^d)$ are such that $Q_0 \ll Q_1$. For $p \in [0,1]$, consider the mixture $Q_p := pQ_0 + (1-p)Q_1$. Then*

$$
C_{LS}(Q_p) \leq \max \big\{ (1 + (1-p)\lambda_p) C_{LS}(Q_0), (1 + p\lambda_p(1 + \chi^2(Q_0, Q_1))) C_{LS}(Q_1) \big\},
$$

*for*

$$
\lambda_p := \begin{cases} \frac{\log p - \log(1-p)}{2p-1} & p \in [0, \frac{1}{2}) \cup (\frac{1}{2}, 1] \\ 2 & p = \frac{1}{2}. \end{cases}
$$

With this result in hand, we just need to get control on the the log-Sobolev constant of $u_m$. This is accomplished in the following lemma.

**Lemma 26** (Log-Sobolev constant of $u_m$) *We have*

$$
C_{LS}(u_m) \leq 16m^2.
$$

*Proof of Lemma 26.* We will apply the Holley-Stroock perturbation principle to a custom comparison distribution. Indeed, for a parameter $r > 0$ to be chosen later, let

$$
\phi_r(x) := \begin{cases} \frac{1}{2}(x-2m)^2 & x > 2m + r \\ \alpha_r\big(x - \frac{m}{2}\big)^2 + \delta_r & x \in [-m-r, 2m+r] \\ \frac{1}{2}(x+m)^2 & x < -m - r, \end{cases}
$$

where we take $\alpha_r, \delta_r$ to make $\phi$ continuously differentiable; in particular

$$
\alpha_r := \frac{r}{2r+3m}, \qquad \delta_r := -\frac{3mr^2}{8}.
$$

Note that $\phi_r$ is $\alpha_r$ strongly convex, so that $C_{LS}(e^{-\phi_r}) \leq \frac{1}{\alpha_r}$. On the other hand

$$
\delta_r - \frac{1}{2}r^2 \leq \phi_r - \frac{1}{2}d(x, I_m)^2 \leq \alpha_r\big(\frac{3m}{2} + r\big)^2 + \delta_r.
$$

Hence

$$\operatorname{osc}(\phi - \frac{1}{2}d(\cdot, I_m)^2) \leq \alpha_r \left(\frac{3m}{2} + r\right)^2 + \frac{r^2}{2}.$$

Hence by the Holley-Stroock perturbation principle Holley and Stroock (1987) we know that

$$C_{LS}(u_m) \leq \exp\left(\alpha_r \left(\frac{3m}{2} + r\right)^2 + \frac{r^2}{2}\right) C_{LS}(e^{-\phi_r}) \leq \frac{1}{\alpha_r} \cdot \exp\left(\alpha_r \left(\frac{3m}{2} + r\right)^2 + \frac{r^2}{2}\right).$$

Taking $r = 1/m$, we find

$$C_{LS}(u_m) \leq (2 + 3m^2) \exp\left(\frac{1}{3m^2}(2m)^2 + \frac{1}{2m^2}\right) = (2 + 3m^2) \exp\left(\frac{4}{3} + \frac{1}{2m^2}\right) \leq 16m^2.$$

$\square$

With the result in hand, we can prove Lemma 24.

*Proof of Lemma 24.* We first compute $\chi^2(\mathcal{N}(m, 1), u_m)$. For this, let us write $u_m = c_{u_m} e^{-\frac{1}{2}d^2(x, I_m)}$, and note that

$$c_{u_m}^{-1} = \int_{\infty}^{\infty} e^{-\frac{1}{2}d^2(x, I_m)} \mathrm{d}x = \sqrt{2\pi} + 3m \leq 4m.$$

Hence

$$\chi^2(\mathcal{N}(m, 1), u_m) + 1 \leq \frac{2m}{\pi} \int_{-\infty}^{\infty} e^{-(x-m)^2 + \frac{1}{2}d(x, I_m)^2} \mathrm{d}x$$

$$= \frac{2m}{\pi} \int_{-\infty}^{-m} e^{-(x-m)^2 + \frac{1}{2}d(x, I_m)^2} \mathrm{d}x + \frac{2m}{\pi} \int_{-m}^{2m} e^{-(x-m)^2 + \frac{1}{2}d(x, I_m)^2} \mathrm{d}x$$

$$+ \frac{2m}{\pi} \int_{2m}^{\infty} e^{-(x-m)^2 + \frac{1}{2}d(x, I_m)^2} \mathrm{d}x$$

$$= \frac{2m}{\pi} \int_{-\infty}^{-m} e^{-(x-m)^2 + \frac{1}{2}(x+m)^2} \mathrm{d}x + \frac{2m}{\pi} \int_{-m}^{2m} e^{-(x-m)^2} \mathrm{d}x$$

$$+ \frac{2m}{\pi} \int_{2m}^{\infty} e^{-(x-m)^2 + \frac{1}{2}(x-2m)^2} \mathrm{d}x$$

$$\leq \frac{2m}{\pi} \int_{-\infty}^{-m} e^{-\frac{1}{2}(x-m)^2} \mathrm{d}x + \frac{2m}{\pi} \int_{-\infty}^{\infty} e^{-(x-m)^2} \mathrm{d}x$$

$$+ \frac{2m}{\pi} \int_{2m}^{\infty} e^{-\frac{1}{2}(x-m)^2} \mathrm{d}x$$

$$\leq \frac{2m}{\pi}\left(\sqrt{2\pi} + \sqrt{\pi} + \sqrt{2\pi}\right) \leq 5m.$$

Using Lemma 26 and plugging into Theorem 25, with $p = 1 - e^{-a^2 m^2/2}$, we find

$$C_{LS}(\pi) \leq \max\left\{1 + (1-p)\lambda_p, 80m^3(1 + p\lambda_p)\right\} \leq 162m^3\lambda_p.$$

The result follows. $\square$

## E  ADDITIONAL NUMERICAL ILLUSTRATIONS

The geometric path is often illustrated in a setup where the initialization is chosen in the middle of a two symmetric modes: see for example Cabezas et al. (2023, Sampling Book, Tempered SMC) or Maurais and Marzouk (2024, Fig. 1) or Chehab et al. (2024, Fig. 1). In this very specific setting, the geometric path conveys a sense that it evolves particle positions. In a more general setting, we can observe in Figure 4 that once the closer mode is reached, the path seems to evolve the particle weights, which is problematic for Langevin dynamics.

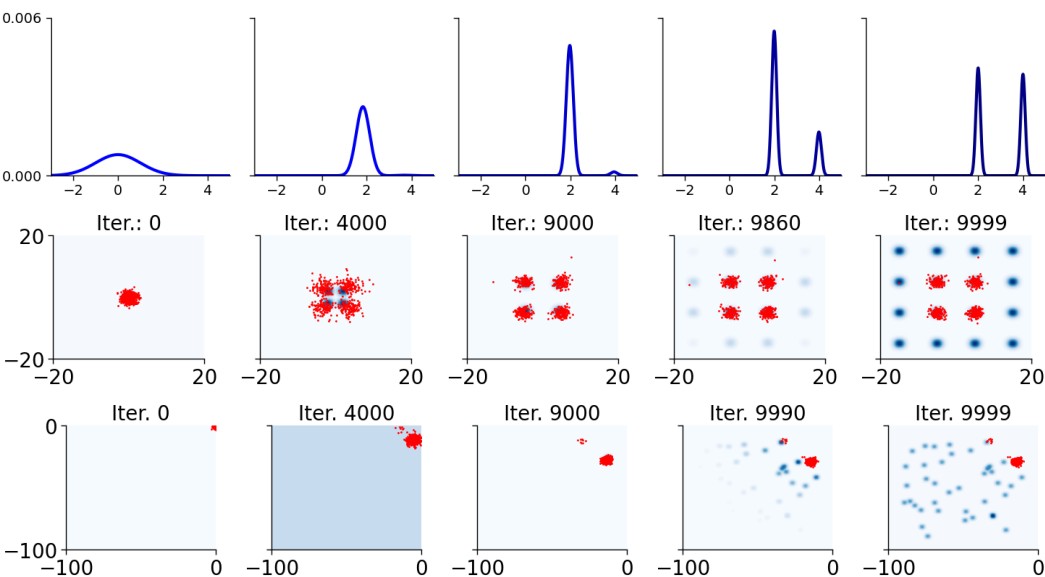

Figure 4: Geometric path from a Gaussian to a Gaussian mixture. We observe that that this path displaces mass "horizontally" to the nearest modes (left columns), and then "vertically" to the remaining modes (right columns). Intuitively, this second part is problematic for a Langevin sampler.

