# OpenReview forum: "Provable Convergence and Limitations of Geometric Tempering for Langevin Dynamics"
_ICLR.cc/2025/Conference — ICLR 2025 Poster_

### Official Review · Reviewer_mTXb · 2024-11-03

**Soundness:** 4
**Presentation:** 4
**Contribution:** 3
**Rating:** 8
**Confidence:** 4

**Summary:**

This work analyzes the convergence rate of Langevin dynamics with geometric tempering (LD-GT), a modification of the Langevin dynamics which attempts to follow the geometric path between a proposal distribution $\nu$ (e.g. a standard Gaussian) and the target distribution $\pi$.
More precisely, LD-GT with tempering schedule $(\lambda_k)_k \subset [0,1]$ is

$$X_0 \sim \nu, ~~~~ X_{k+1} = X_k + h \nabla \log \mu_k(X_k) + \sqrt{2 h} \epsilon_k,$$

where $\mu_k \propto \nu^{1-\lambda_k} \pi^{\lambda_k}$ is the geometric path (and $\epsilon_k$ are independent Gaussians).

(Per the authors' account of the literature,) LD-GT was proposed since the 1990s, and one of the motivations is the intuition that sampling progressively from the path $\mu_k$ is easier than sampling directly from the target $\pi$, especially if $\pi$ is multi-modal.

This work's contributions are two-fold:
- Precise convergence guarantees for LD-GT under certain common assumptions on the proposal $\nu$ and the target $\pi$: Poincare inequality (PI), log-Sobolev inequality (LSI), strong log-concavity.\
To this aim, a key sub-question is to estimate the PI or LSI constant of the path $(\mu_k)_k$. Besides showing how to best utilize well-known upper bounds, the authors also identify cases where these constants are surprisingly poor, leading to the next item.
- This work provides evidence that the original intuition motivating LD-GT is wrong, by exhibiting cases where LD-GT must converge very slowly (regardless of the choice of schedule). Remarkably this can even happen for well-conditioned and uni-modal targets $\pi$, for which vanilla LD can be expected to converge fast.

**Strengths:**

This paper addresses a natural question on the convergence of a sampling algorithm. The positive results (first item in "Summary") are of theoretical interest, as they address the technically difficult question of optimizing the upper bounds w.r.t the temperature schedule. The negative findings are surprising: namely, adding geometric tempering may actually slow down Langevin dynamics. This new insight is significant for both theory and practice.

The presentation is very clear and "flows" very nicely. All the technical claims are correct are far as I checked.

**Weaknesses:**

No substantial weaknesses, but the negative results of this paper naturally lead to a question which is not addressed nor mentioned in this paper, see "Questions" below.

Minor comments on the presentation:
- use citep instead of citet on lines 119, 233, 254, 468
- correct typos and/or grammar on lines 271, 326, 420, 496, 1389, 1494, 1838
- justify the fact that chi^2, KL > TV rather than say it "of course" holds (line 493)
- line 988 contains the proof of Corollary 13, not 17
- add details on the argument on line 1824 (I could not reconstruct it using Cauchy-Schwarz, only Jensen)
- use different markers for each curve in Figure 2
- consider using a log scale or showing less iterations in Figure 3
- consider including the example of section 4.1 in Figure 4, in addition to Figure 1 (which shows only $\lambda \in \{0, 0.45, 1\}$)

**Questions:**

The theoretical results in this work suggest that geometric tempering may not help the convergence of Langevin dynamics. Yet tempering is a strategy that is used in practice (per the authors' presentation of the literature). In practice, is tempering observed to lead to improved performance compared to vanilla Langevin dynamics? If yes, is there any intuitive reason why?

Minor questions:
- Would the conclusions, and the analysis techniques, of this paper still apply if one takes $\nu$ to be the Lebesgue measure instead of a probability measure?
- In practice, is the proposal $\nu$ always taken to be a Gaussian? Is it sometimes taken to be the Lebesgue measure? Or multi-modal?
- Is the $\frac{1}{2\alpha_\pi t}$ rate in Proposition 7 (line 433) classical? If so please give a reference.

---

> ### Author Response · Authors · 2024-11-19
> **Response to Reviewer mTXb**
>
> We thank reviewer mTXb for their detailed reading of our submission, especially the appendix. We have integrated all their suggestions relating to presentation and typos.  We respond to the reviewer's question
> about how practical successes of tempering mesh with our theory
> in the general response above, and here respond to the
> minor questions.
>
> **Varying the proposal distribution $\nu$ ---**
> For our lower bounds, we use the proposal $\nu = \mathcal{N}(0, 1)$
> as is common [1, 2, 3] (indeed this choice
> is even a default in the Bayesian software package Blackjax),
> and the analysis is indeed tailored to this choice. The other common
> choice, which you mention,
> is the so-called ``uniform proposal",
> where $\nu = \mathcal{L}$ is the Lebesgue measure.
> This setting, however, is not precisely covered by our
> framework, since we need $\nu$ to be a probability measure.
> On the other hand,
> our upper bounds can nearly be extended to this case by using the following trick. Consider the
> scheme $\pi^{\gamma_t}$, for $\gamma_0 > 0$ and $\gamma_t$
> non-decreasing.
> Then if we let $\nu = \pi^{\gamma_0}$
> and set $\lambda_t := \frac{\gamma_t - \gamma_0}{1 - \gamma_0}$,
> we obtain $\mu_t = \pi^{\gamma_t}$. In other words,
> we can recover the uniform proposal in our framework
> so long as $\gamma_0 > 0$,
> so that there is initially at least some weight on $\pi$.
>
>
> **Convergence rate of tempering with a linear schedule in Proposition 7 ---** We are not aware of any references
> which establish similar rates as this result, either for tempered
> or non-tempered Langevin.
>
>
> [1] Zhang et al. Differentiable Annealed Importance Sampling and the Perils of Gradient Noise. NeurIPS, 2021.
>
> [2] Thin et al. Monte Carlo Variational Auto-Encoders. ICML, 2021.
>
> [3] Dai et al. An Invitation to Sequential Monte Carlo Samplers. Journal of the American Statistical Association, 2020.

---

> > ### Comment · Reviewer_mTXb · 2024-11-21
> >
> > Thank you for your answer.
> >
> > That's a nice trick for covering the uniform proposal! I think it could be worth adding as a remark in the paper. You say your bounds can "nearly" be extended using this trick; why "nearly"?
> >
> > Regarding Proposition 7, I would suggest removing or rephrasing the sentence on line 433, as the word "recover" can be interpreted to mean that this rate is well-known.

---

> > > ### Author Response · Authors · 2024-11-21
> > > **Answer to Reviewer mTXb**
> > >
> > > Thank you! We will add this trick to the paper: it "nearly" covers the uniform case because it relies on the assumption that  $\lambda_0 > 0$. This is a reasonable assumption in practice, but not an exhaustive one in theory, where one could consider tempering schedules which do start at $0$.
> > >
> > > We agree with the wording issue around "recovers" and will modify. Thank you for picking up on this!

---

### Official Review · Reviewer_pomC · 2024-11-04

**Soundness:** 4
**Presentation:** 3
**Contribution:** 3
**Rating:** 8
**Confidence:** 3

**Summary:**

This paper offers a thorough study of geometric tempering combined with a Langevin MCMC scheme. In particular, a general theory is given which characterizes the error induced by said dynamics for arbitrary tempering schemes. Negative results are then given for the efficacy of tempering schemes (over the naive Langevin dynamics) both in terms of the intermediate distributions' log-Sobolev constants, as well as the worst-case convergence rate, although some regimes where the tempering is beneficial are highlighted.

**Strengths:**

The main positive result on geometric tempering (Theorem 3) seems quite thorough (in that it comprises every reasonable regime of interest) and about as good as one could hope for in this context.

The negative example is very intuitive and is a worthwhile inclusion into the paper. It offers a good characterization about why one might be skeptical about the occasional poor performance of these schemes in practice, and gives good intuition about the heart of the problem (the appearance of multimodality).

The inclusion of more concrete lower bounds is also insightful.

**Weaknesses:**

It would be more helpful if the paper offered more positive examples of instances where the tempering can improve over vanilla Langevin by at least polynomial factors; in particular, a comparative bound would be helpful in Propositions 6, 7.

It would also be good if the paper could explore further the areas where tempering has a provable benefit over Langevin, especially in cases of multimodality where the algorithm would likely be used.

**Questions:**

The following suggestions relate to minor areas of the paper:

In Figure 2, should we not be scaling the $y$-axis logarithmically for a more reasonable demonstration?

The comment after (3) is strange. Probably, you mean to take $kh = t$ for a fixed choice of $t \in \mathbb R$, and then some schedule $h = t/K$ for a set of integers $K$, rather than what is written.

There is a spacing issue in Line 190~191.

Line 240: Lebesgue -> Lebesgue measure.

Line 425: where obtain-> where we obtain

It is a bit strange to cite Durmus 2019 for the Langevin rate in the str. convex + smooth setting, compared to earlier work such as [1].

Durmus, Alain, and Eric Moulines. "High-dimensional Bayesian inference via the unadjusted Langevin algorithm." (2019): 2854-2882.

---

> ### Author Response · Authors · 2024-11-19
> **Response to Reviewer pomC**
>
> We thank reviewer pomC for their attentive reading of our paper and positive review. We have integrated all their suggestions relating to presentation and typos. We agree with reviewer pomC that discussing when tempering can be useful in practice, as well
> as comparing to vanilla Langevin is important, and do so in our general response to all reviewers.

---

### Official Review · Reviewer_8HQm · 2024-11-05

**Soundness:** 3
**Presentation:** 3
**Contribution:** 3
**Rating:** 6
**Confidence:** 4

**Summary:**

This paper presents a theoretical analysis of geometric tempering when applied to Langevin dynamics, a popular sampling method in machine learning and statistics. Geometric tempering is a technique that attempts to improve sampling from complex multi-modal distributions by sampling from a sequence of intermediate distributions that interpolate between an easy-to-sample proposal distribution and the target distribution. The authors provide the first convergence analysis under functional inequalities, proving both upper and lower bounds for tempered Langevin dynamics in continuous and discrete time. They also derive optimal tempering schedules for certain pairs of proposal and target distributions.

**Strengths:**

Perhaps surprisingly, the paper's findings are largely negative regarding the effectiveness of geometric tempering. The authors demonstrate that geometric tempering can actually worsen functional inequalities exponentially, even when both the proposal and target distributions have favorable properties. Through theoretical analysis, they show a simple bimodal case where geometric tempering takes exponential time to converge. More strikingly, they prove that similar poor convergence results can occur even with unimodal target distributions that have good functional inequalities. These results suggest that geometric tempering may not only fail to help with convergence but could actually be harmful in some cases, challenging the conventional wisdom about its utility.

**Weaknesses:**

In this paper they consider targets of the form $\nu^{1 - \lambda} \pi^{\lambda}$, where $\nu$ is called the proposal. In many other prior works, the targets are of the form $\pi^{\lambda}$, which corresponds to $\nu$ being an improper uniform distribution. This seems to be the main source of the largely negative results provided in this paper. Could the authors clarify the reason for considering target the above form?

**Questions:**

Please see question abobe

---

> ### Author Response · Authors · 2024-11-19
> **Response to Reviewer 8HQm**
>
> We thank reviewer 8HQm for their careful reading of our paper.
> Let us respond to their question concerning the motivation
> for considering Gaussian proposals
> $\nu = \mathcal{N}(0, 1)$.
> We emphasize that these proposals are indeed used
> in practice, especially in Bayesian
> settings. For example, see (1, 2, 3)
> for three papers which use Gaussian proposals. In fact,
> the popular Bayesian software library Blackjax even makes the Gaussian
> proposal the default initialization.
> More generally, geometric tempering makes sense for a broad variety
> of proposal distributions, and so we believe that an investigation
> at this level of generality is of basic interest.
> Finally, we mention that our upper bounds can yield bounds for schemes
> of the form $\pi^{\gamma_t}$, so long as $\gamma_0 > 0$ and is
> non-decreasing:
> indeed, if we let $\nu = \pi^{\gamma_0}$
> and set $\lambda_t := \frac{\gamma_t - \gamma_0}{1 - \gamma_0}$,
> then $\mu_t = \pi^{\gamma}$.
>
> [1] Zhang et al. Differentiable Annealed Importance Sampling and the Perils of Gradient Noise. NeurIPS, 2021.
>
> [2] Thin et al. Monte Carlo Variational Auto-Encoders. ICML, 2021.
>
> [3] Dai et al. An Invitation to Sequential Monte Carlo Samplers. Journal of the American Statistical Association, 2020.

---

> > ### Comment · Reviewer_8HQm · 2024-11-25
> > **Response to Authors**
> >
> > Thanks for the clarification. I maintain my score and I have no objections to the paper being accepted.

---

> > > ### Author Response · Authors · 2024-11-27
> > > **Response to Reviewer 8HQm**
> > >
> > > We thank the reviewer: are there any remaining concerns that could be addressed to update the score?

---

### Official Review · Reviewer_Exbp · 2024-11-08

**Soundness:** 3
**Presentation:** 3
**Contribution:** 3
**Rating:** 6
**Confidence:** 3

**Summary:**

This work studies the convergence guarantee of geometric tempering for the Langevin diffusion and its time-discretization the Langevin algorithm. The authors prove a convergence rate under a general tempering schedule, demonstrating dependency on the isoperimetry of the intermediate probability measures, in particular their log-Sobolev constant. While this constant can be suitably controlled when both measures are strongly log-concave, the authors show that even when both proposal and target densities are unimodal, intermediate measures can suffer from a poor log-Sobolev constant that scales exponentially with the distance between the modes of proposal and target measures.

**Strengths:**

While I am not an expert in annealing or tempering algorithms for sampling, it seems that this is the first paper that proves the convergence of geometric tempering for the Langevin diffusion using functional inequalities, which is interesting. The negative results also provide a good example of why tempering may not work in practice, despite both proposal and target measures having suitable isoperimetry.

**Weaknesses:**

* The related work section could be better structured. For example, breaking into multiple paragraphs and adding paragraph titles could help with readability and following the discussion.
* The lower bound examples hold in dimension 1, and show exponentially bad dependence on the distance between modes. While these bounds are interesting, it is not very intuitive to me why it would be natural for the distance between modes to grow in fixed dimensions. On the other hand, in a high-dimensional setting, it is more intuitive that $m$ grows with square root of dimension. Could it be straightforward to (perhaps only intuitively) extend the lower bounds to high-dimensional settings?

**Questions:**

* I believe for $KL(p_0, \mu_0)$ to disappear in Corollary 5, one needs to set $\lambda_0 = 0$. In that case, it would not be possible to choose $\lambda_t = 1$ for all $t > 0$ in a continuous manner.

* If all $\lambda_i$s are very close to 1 in Theorem 9, we are effectively running vanilla Langevin. In that case, why should we have exponential convergence time?

* The vanilla Langevin analysis only requires the log-Sobolev inequality and smoothness for discretization. Why do we additionally need dissipativity of proposal and target measures here?

    * In fact, the Langevin algorithm is known to convergence under extremely mild conditions, namely a weak Poincaré inequality (which holds for all locally bounded potentials, although without explicit control on the constant) and smoothness of the gradients, see e.g. [1] and references therein. Are there major challenges for obtaining convergence guarantees under (weak) Poincaré inequalities for the tempered Langevin algorithm?

* Is there a sense in which one can choose optimal proposal distributions $\nu$ when we only know some information about $\pi$?

* I believe a summation over $i$ is missing in Equation (12).

* Some typos:
    * Line 152 missing absolute continuity before “... and $+\infty$ otherwise”.
    * Line 175: potential -> potentials
    * Line 119, 233, 253, 467: missing parentheses in citation
    * Line 238: satisfy -> satisfies
    * Line 336, 339, 383: section … -> Section …
    * Line 420: are unknown -> is unknown
    * Line 425: where we obtain
    * A typo in Line 439 makes the sentence unreadable.




---
[1] A. Mousavi-Hosseini, T. Farghly, Y. He, K. Balasubramanian, M. A. Erdogdu. "Towards a Complete Analysis of Langevin Monte Carlo: Beyond Poincaré Inequality". COLT 2023.

---

> ### Author Response · Authors · 2024-11-19
> **Response to Reviewer Exbp**
>
> We thank the reviewer Exbp for their attentive reading: we will correct all typos and better structure the related works paragraphs.
> We respond below to their other questions and comments.
>
> **Applicability of Corollary 5 ---** the corollary requires initializing Langevin dynamics at the proposal, i.e. $p_0 = \nu$,
> but we do not require that $\lambda_0 = 0$.
> We refer the reviewer to the short proof in Appendix C.1:
> to control $\mathrm{KL}(p_0\|\mu_0)$ we use the tempering rule,
> and ultimately simply bound $\lambda_0$ by $1$.
>
> **Theorem 9 in case $\lambda_i$ is close to $1$---**
> We are in complete agreement with the reviewers' intution,
> and note that in Theorem 9, $\delta_k$ is defined as $\delta_k := 8m^2 e^{-(1 - \lambda_k)m^2}$,
> so that when $\lambda_k$ is close to $1$, there is no exponential
> slow-down. Actually, Theorem 9 makes this intuition quantitative
> and states that so long as $1 - \lambda_k \leqslant C m^{-2}$,
> the slow-down will only be of polynomial order in the mean separation.
>
>
> **Choosing an optimal proposal distribution knowing some information about the target distribution ---** consider the setup when the target is a mixture of two symmetric Gaussian distributions and we initialize with a Gaussian located “in between” the two target modes, specifically at the barycenter. Then, convergence is provably fast for Tempered Langevin [1, Example 1] and related sampling processes [2]. Yet, initializing in this way requires knowing the locations of all target modes: this would require solving  a “global optimization” problem, which can be a harder problem than the original problem of sampling from the target distribution [3].
>
> **Do the lower bounds hold in higher dimensions? ---**
> We expect that the lower bounds go through in higher dimensions
> with minor modifications. The key high-level point in all
> of the lower bounds is that at intermediate times,
> the tempering path becomes bimodal, and if the
> law $p_t$ of the particle following the tempering
> is too concentrated in one of the modes vs. another,
> it will take exponential time to spread mass between the modes
> (for example, see Prop. 22 in Appendix D).
> In other words, and as illustrated in Fig. 4 Appendix E, the core phenomenon behind our lower bounds is the fact that along the geometric path, mass tends to “teleport” from one mode to another,  preventing Langevin to converge as the particles eventually get stuck in the first encountered mode.
>
> All of this translates intuitively, and likely rigorously,
> without issue to higher dimensions.
> We chose to present the lower bounds in dimension $1$
> for simplicity, as well as
> to make it clear that the problem is not a curse of dimensionality (which may be common across methods),
> but instead a problem specifically with the tempered Langevin itself.
>
> **Why additional dissipativity assumption? ---**
> Our only use of the dissipativity assumptions
>  is to control the second moment of $p_t$, the law of the process $X_t$ given in (9).
>  At a technical level, the reason why we have this extra assumption
>  as compared to the standard analyses of vanilla Langevin
>  is precisely because of the additional terms arising from
>  the tempering. Weakening this assumption further
>  is an interesting direction for future work.
>
> **Convergence under weaker functional inequalities
> than log-Sobolev? ---**
> The main technical novelty of our analysis is the
> way that we deal with the new terms
> arising from the tempering dynamics (see Step 1 and Step 2 in Appendix A.2, as well as the supporting Lemmas in Appendix A.4).
> In particular,
> the extra terms which arise here are particularly suitable
> to analysis when the Lyapunov function is $\mathrm{KL}$.
> For example, we are not
> aware of a straightforward means of extending
> our analysis to $\chi^2$.
> Since these alternative functional inequalities imply
> convergence in alternative Lyapunov functions (e.g. Poincaré involves
> analysis in $\chi^2$), we are therefore not aware of a straightforward
> extension of our results to weaker isoperimetric assumptions.
>
>
> [1] Guo et al. Provable Benefit of Annealed Langevin Monte Carlo for Non-log-concave Sampling. Arxiv, 2024.
>
> [2] Madras and Zheng. On the Swapping Algorithm.
> Journal of Random Struct. Algorithms, 2003.
>
> [3] Ma et al. Sampling can be faster than optimization. PNAS, 2019.

---

> > ### Author Response · Authors · 2024-11-27
> > **Follow-up Response to Reviewer Exbp**
> >
> > We hope to have addressed all the reviewer's concerns, please let us know if there is anything we can further clarify.

---

> ### Comment · Reviewer_Exbp · 2024-11-29
>
> I thank the authors for their detailed response. I'm happy to recommend acceptance, and have decided to keep my score based on my perceived significance of the results.

---

### Author Response · Authors · 2024-11-19
**Response to all reviewers**

We thank all four reviewers for their positive feedback. In this general reply, we address a question that was
shared among reviewers. The remaining comments
are addressed in detail in the individual replies.


**When does tempering outperform Langevin? (reviewers pomC and mTXb) ---**
Reviewer pomC asked us to identify settings where
tempering outperforms Langevin. And reviewer mTXb asked
how our results fit with the fact that
tempering is often used successfully in practice.
For both of these questions, we refer to our upper bounds,
Theorems 1 and 3. These results gives rates of convergence
for tempered Langevin dynamics in terms of the inverse log-Sobolev constants
$\alpha_t$ along the tempering path. While it is true that
our lower bound on the log-Sobolev constant in Theorem 4
rules out the possibility of tempering generically improving the log-Sobolev
constant, we emphasize that Theorem 4 only shows
one poorly conditioned example, specific to a pair of proposal and target distributions.
In particular, in any given example,
it may happen that the intermediate log-Sobolev contants $\alpha_t$
are significantly better than those of the target $\pi$ (this is, essentially,
the core intuition behind tempering).
In such a case, tempering can be expected to converge more quickly
than vanilla Langevin.

To gain some intuition for why this can happen,
imagine $\pi$ is a bimodal distribution, $\nu$ is a wide distribution
roughly uniformly spread over the modes,
and the initial-time distribution $p_0$ is concentrated in one mode.
In general, the log-Sobolev constant $\mu_t$ should scale
as the height of the energy barrier between the modes: this is the hill
that a particle must cross to move from one mode to the other.
But, when $\lambda_t$ is small, the target $\mu_t \approx \pi^{\lambda_t}$,
so, in particular,
the height of the energy barrier is significantly smaller.
Thus, the log-Sobolev constants of $\mu_t$ should be significantly better
than those of the target, early in the tempering.

Given such control of the log-Sobolev constants, we could
plug it in to Theorem 1 to obtain a rate with which could
then improve on vanilla Langevin.
Rigorously obtaining such control is a fascinating open question which builds upon
the theory developed in this work;
we expect that this will delicately depend on the details
of the specific example in consideration. Generally speaking,
the control of log-Sobolev constants for mixtures and other multimodal distribution
is a challenging area of ongoing research,
see [1, 2] for some recent
work
in this direction.
To finally answer reviewer mTXb's question:
situations where tempering performs well in practice
can be explained within our theory
as situations where the intermediate log-Sobolev constants
are significantly better than those of the target.
And to respond to reviewer pomC:
our upper bounds provide a general framework, through
the log-Sobolev constants, to understand when
tempering does and does not improve on vanilla Langevin.
Although Theorem 4 (and our other lower bounds) rules out generic good behavior of the tempering,
there still exist
situations (such as in Figure 2) in practice where the behavior is better than that
of vanilla Langevin,
and our results
provide a solid foundation for future research into this phenomenon.

[1] Chen, Hong-Bin, Sinho Chewi, and Jonathan Niles-Weed. ``Dimension-free log-Sobolev inequalities for mixture distributions." Journal of Functional Analysis 281.11 (2021): 109236.

[2] Schlichting, André. ``Poincaré and log–sobolev inequalities for mixtures." Entropy 21.1 (2019): 89.

---

### Meta-Review · Area_Chair_a5tY · 2024-12-17

**Metareview:**

The authors analyze the convergence of tempered Langevin dynamics. Most interestingly, the authors derive lower bounds where tempering leads to exponentially poor convergence. This result is quite novel as agreed upon by all reviewers. On top of this, the authors also provide the first upper bound results on convergence under functional inequality, which is already a nice result in itself.

No significant criticisms were raised during the review process, and both reviewers with a score of 6 have agreed with acceptance, albeit not good enough to raise the scores to 8. Therefore, I believe this paper is welcomed contribution to the sampling literature and a clear accept.

**Additional Comments On Reviewer Discussion:**

A common theme among all the reviewers is that everyone is impressed by the negative result on tempering. This seems to be a genuinely novel and surprising result, since the target distributions appear to satisfy nice functional inequalities.

---

### Decision · Program_Chairs · 2025-01-22

Accept (Poster)